# Parameter Manifold Purification

**Jiacong Hu** [1 2]  **Jinxun Wu** [1 2]  **Shengxuming Zhang** [1 2]
**Shunyu Liu** [3]  **Haofei Zhang** [1 2]  **Mingli Song** [1 2]  **Zunlei Feng** [4]

## Abstract

Deep models are vulnerable to performance degradation caused by various factors, such as imbalanced samples, inaccurate labels, and backdoor attacks. However, existing optimization methods that address these issues are typically designed in a scenario- or architecture-specific manner, and each optimization often requires costly training. To this end, inspired by image denoising, we propose *parameter purification* as a new paradigm for model performance optimization. Parameter purification attributes performance degradation to the contamination of model parameters and aims to recover clean parameters from corrupted ones in a manner analogous to image denoising. To purify parameters with massive scale and complex structure, we further introduce a novel *parameter manifold purification* method. In this framework, high-dimensional and complex parameters are first viewed as manifolds embedded in a high-dimensional space, and are then partitioned into nested local parameter-cluster manifolds via a proposed parameter clustering strategy. Meanwhile, to remove parameter redundancy while preserving global parameter information, we propose an implicit manifold autoencoder along with a parameter-cluster discrepancy loss to learn low-dimensional representations of parameter-cluster manifolds. Finally, an implicit conditional diffusion model is applied to denoise the low-dimensional parameter manifolds, progressively restoring clean parameters. Extensive experiments under three representative scenarios that cause model performance degradation demonstrate that parameter manifold purification can accurately and completely purify cor-

[1]State Key Laboratory of Blockchain and Data Security, Zhejiang University [2]Hangzhou High-Tech Zone (Binjiang) Institute of Blockchain and Data Security [3]Nanyang Technological University [4]School of Software Technology, Zhejiang University. Correspondence to: Zunlei Feng <zunleifeng@zju.edu.cn>.

*Proceedings of the $43^{rd}$ International Conference on Machine Learning*, Seoul, South Korea. PMLR 306, 2026. Copyright 2026 by the author(s).

rupted parameters of unseen models, analogous to denoising unseen images, and rapidly improve model performance. Project page: https://parameter-generation.github.io/.

## 1. Introduction

As carefully engineered yet highly complex systems, deep models are vulnerable to performance degradation caused by various adverse factors, such as imbalanced samples (Huang et al., 2016; Cao et al., 2019; Zhang et al., 2023), inaccurate labels (Natarajan et al., 2013; Li et al., 2020; Song et al., 2022), and backdoor attacks (Liu et al., 2020b; Li et al., 2021c; 2022c). Although numerous approaches have been proposed to address these issues, including learning under imbalanced samples, learning with noisy labels, and backdoor defense (Liu et al., 2022; Tian et al., 2022; Huang et al., 2022), existing methods either rely on strong prior and are thus limited to specific scenarios or network architectures (Liu et al., 2018a; Huang et al., 2022), or require costly training to obtain a usable model each time optimization is needed (Patrini et al., 2017; Zhu et al., 2022).

To address this pervasive and urgent problem, we draw inspiration from image denoising (Kawar et al., 2022; Kulikov et al., 2023; Zhu et al., 2023) and propose a novel paradigm termed *parameter purification*. Parameter purification aims to serve as a general optimization framework that directly improves degraded models without scenario or architecture constraints, and without requiring training of the corrupted model during optimization. Specifically, analogous to noisy images, parameter purification attributes model performance degradation to the contamination of model parameters, and seeks to recover clean parameters from corrupted ones in a manner similar to image denoising. However, compared with modeling images, model parameters are far more challenging to handle due to their massive scale and complex structural organization (Frankle & Carbin, 2018).

Despite a growing body of work that explores the properties of model parameters (Schürholt et al., 2021; 2022; 2024) or even generates new parameters (Nava et al., 2022; Wang et al., 2024a; Xie et al.), none of them studies how to directly optimize performance-degraded models through parameter-level purification. Moreover, these methods are

difficult to adapt to the parameter purification setting. The first key challenge lies in the fact that parameter contamination may occur anywhere in a model, which requires parameter modeling with high completeness. In contrast, existing approaches are often limited to simple neural architectures (Ha et al., 2016; Zhang et al., 2018), can only generate a small subset of parameters (Wang et al., 2024a; Soro et al., 2024), or fail to generalize across multiple models (Ashkenazi et al., 2022; Yang & Wang, 2025), making it difficult to purify all parameters of models with diverse architectures. The second challenge is that parameter purification aims to further optimize model performance, which demands highly accurate parameter modeling. However, existing works are primarily designed for predicting parameter attributes (Schürholt et al., 2021; Kofinas et al., 2024) or for adapting models to downstream tasks such as multi-task learning (Requeima et al., 2019; Raychaudhuri et al., 2022) and few-shot learning (Zhang et al., 2024; Nava et al., 2022), lacking effective mechanisms for high-precision parameter learning that can directly improve model performance.

To overcome these challenges, we propose a novel *Parameter Manifold Purification* (PMP) framework that enables complete and accurate parameter purification. Specifically, to handle the massive scale and complex structure of model parameters, we first view all parameters as a manifold embedded in a high-dimensional space. This global parameter manifold is then decomposed in a divide-and-conquer manner via a parameter-cluster-based partitioning strategy. Under this formulation, the entire set of model parameters corresponds to a global manifold in high-dimensional space, while each partitioned parameter cluster forms a local manifold nested within it. To further eliminate the substantial redundancy present in parameter clusters while preserving global parameter information under the divide-and-conquer paradigm, we are inspired by implicit neural representations for modeling complex spaces (Sitzmann et al., 2019; Mildenhall et al., 2021; Chen et al., 2021b) and propose an *implicit manifold auto-encoder*. This module learns compact low-dimensional manifolds for parameter clusters through implicit manifold modeling and the embedding of manifold positional information. In addition, to address the imbalance of parameter magnitudes both within and across clusters, we introduce a *parameter-cluster discrepancy loss* with intra-cluster and inter-cluster weighting, which balances the influence of heterogeneous parameter distributions. Finally, for the learned low-dimensional parameter-cluster manifolds, we employ an implicit conditional diffusion model to perform precise denoising while preserving parameter completeness, thereby purifying corrupted parameters.

The overall PMP framework does not require access to clean data or any additional supervision at test time. Instead, it leverages reference model parameters obtained from standard training procedures in an offline stage to learn a prior over clean parameter spaces, which is then applied to unseen models. Extensive experiments across representative scenarios that contaminate model parameters, including class imbalance, noisy labels, and backdoor attacks, demonstrate that the proposed parameter manifold purification can, analogously to image denoising, rapidly and effectively purify parameters of unseen degraded models and substantially improve their performance. In summary, our main contributions are as follows:

- We introduce a novel task, termed *parameter purification*, which serves as a general optimization paradigm to purify parameters of unseen performance-degraded models in an image-denoising-like manner, directly improving model performance.

- To realize parameter purification, we propose a novel *parameter manifold purification* method, which meets the requirements of high parameter completeness and high precision through a new parameter-cluster manifold partitioning strategy, an implicit manifold auto-encoder, and a parameter-cluster discrepancy loss.

- Extensive experiments on diverse representative scenarios that cause model performance degradation demonstrate that the proposed parameter manifold purification can completely and accurately purify parameters of unseen degraded models, achieving rapid and significant performance improvements.

## 2. Related Works

Achieving parameter purification inevitably requires learning over parameters. Existing studies on parameter learning can be broadly categorized into two major groups: *Hyper-Networks* and *Parameter Diffusion*. Additional details of related works are provided in the *Appendix*.

### 2.1. HyperNetworks

In this paper, HyperNetworks refer to neural networks that generate parameters for other deep neural networks without involving diffusion models. Beginning with HyperNetwork (Ha et al., 2016), an increasing number of related works (Krueger et al., 2017; Sendera et al., 2023; Raychaudhuri et al., 2022; Requeima et al., 2019; Von Oswald et al., 2019) have been proposed to generate parameters for deep architectures. HyperNetworks can be classified by their architectures into MLP-based (Shamsian et al., 2021; Zhao et al., 2020; Shen et al., 2018; Liu et al., 2019b; Navon et al., 2020), CNN-based (Klocek et al., 2019; Nirkin et al., 2021; Alaluf et al., 2022; Brock et al., 2017), GNN-based (Zhang et al., 2018; Knyazev et al., 2021; 2023), sequence-model-based (Ha et al., 2016; Zhmoginov et al., 2022; Volk et al., 2022), and GAN-based generative approaches (Deutsch,

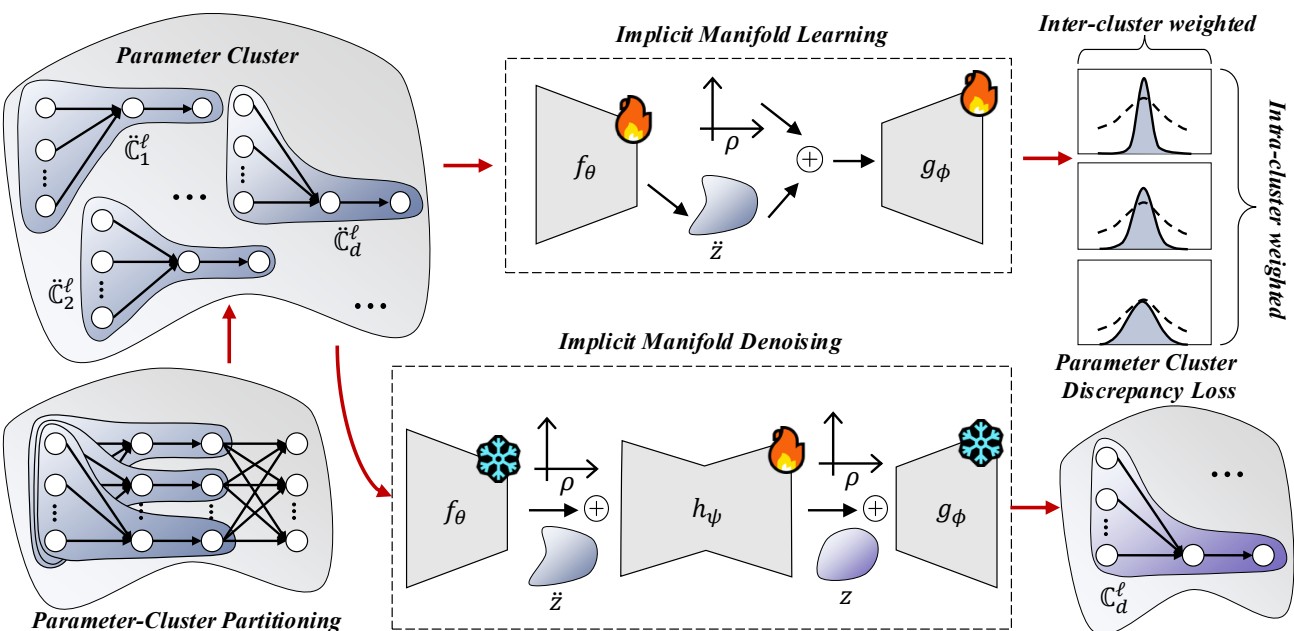

*Figure 1.* Overview of the PMP framework, which includes the core steps of parameter-cluster partitioning, parameter-manifold learning, and parameter-manifold denoising. Symbols without and with the ¨ notation denote contaminated and clean parameters, respectively.

2018; Ratzlaff & Fuxin, 2019). These HyperNetworks have been widely applied to multi-task learning (Raychaudhuri et al., 2022; Requeima et al., 2019; Klocek et al., 2019), few-shot learning (Sendera et al., 2023; Zhao et al., 2020; Zhmoginov et al., 2022), continual learning (Von Oswald et al., 2019), federated learning (Shamsian et al., 2021), model pruning (Liu et al., 2019b), and multi-objective optimization (Navon et al., 2020). Representative methods include the MLP-based centralized HyperNetwork proposed by Shamsian et al. (2021), which generates parameters for each client's small fully connected model in federated learning. Additionally, Zhmoginov et al. (2022) introduced a Transformer-based HyperNetwork to directly generate weights for simple layers in convolutional architectures.

### 2.2. Parameter Diffusion

Parameter Diffusion (Wang et al., 2024a; Saragih et al., 2025; Wang et al., 2025b; Shi et al., 2025; Wang et al., 2025a; Liang et al., 2025; Eyring et al., 2025; Shen et al., 2025; Wang et al., 2025c) refers to approaches that generate parameters using diffusion models (Ho et al., 2020; Nichol & Dhariwal, 2021; Croitoru et al., 2023). These approaches can be grouped according to how parameters are learned: directly diffusing parameters (Xie et al.; Rangel et al., 2024; Du et al., 2023; Zhang et al., 2024; Lutati & Wolf, 2023; Peebles et al., 2022; Yuan et al., 2024; Li et al., 2024) or diffusing parameter representations (Wang et al., 2024a; Wei et al., 2024; Soro et al., 2024; Nava et al., 2022; Lin et al., 2024; Wu et al., 2024; Jin et al., 2024). Their structural organization can be categorized into cross-layer concatenation learning (Wang et al., 2024a; Wei et al., 2024;

Wu et al., 2024; Jin et al., 2024), per-layer learning (Xie et al.; Rangel et al., 2024; Zhang et al., 2024; Lutati & Wolf, 2023; Peebles et al., 2022), and random block partition learning (Soro et al., 2024; Yuan et al., 2024; Li et al., 2024). In terms of application, these methods support pure parameter generation (Wang et al., 2024a; Wei et al., 2024; Lin et al., 2024), few-shot learning (Soro et al., 2024; Nava et al., 2022; Zhang et al., 2024; Yuan et al., 2024), prototype learning (Du et al., 2023), unlearning (Rangel et al., 2024), and more. For instance, Wang et al. (2024a) demonstrated that unconditional diffusion over concatenated subsets of parameters can effectively model parameter distributions. Furthermore, Soro et al. (2024) proposed a conditional diffusion approach based on intra-layer block partitioning to generate parameters for a small number of layers, enabling improved few-shot learning.

In summary, neither HyperNetworks nor Parameter Diffusion methods study a general optimization paradigm that addresses parameter *contamination* caused by diverse adverse scenarios, particularly from the perspective of parameter purification. Moreover, current methods lack effective mechanisms applicable to parameter purification, especially when facing the challenges of high completeness and high precision required for purifying contaminated parameters. Due to space limitations, additional distinctions between our work and prior studies are provided in the *Appendix*.

### 3. Parameter Manifold Purification

Parameter Manifold Purification consists of three key steps: parameter-cluster partitioning, parameter manifold learning,

and parameter manifold denoising, as illustrated in Fig. 1.

## 3.1. Parameter-Cluster Partitioning

Modern deep models contain massive and highly complex parameters, making direct purification of all parameters impractical. To this end, we treat the parameters of large-scale deep models as lying on a high-dimensional manifold, and follow a divide-and-conquer strategy by partitioning parameters before purification. Although partition-based learning has been adopted in several parameter-learning studies (Jin et al., 2024; Lutati & Wolf, 2023; Yuan et al., 2024), existing approaches either partition at too coarse a granularity, making them still unsuitable for large models (Jin et al., 2024; Wu et al., 2024), or partition in irregular patterns that lose structural information, preventing full modeling of the parameters (Yuan et al., 2024; Lutati & Wolf, 2023).

Inspired by interpretability studies suggesting that deep models consist of different functional modules (Geva et al., 2020; Hu et al., 2024b), such as findings showing that individual convolution kernels in CNNs capture distinct factors like color, shape, and texture (Erhan et al., 2009; Mahendran & Vedaldi, 2015; Feng et al., 2022), and that different neurons in Transformers encode and store different types of information (Hu et al., 2024a; Voita et al., 2019; Geva et al., 2022; 2020), we propose a parameter partitioning strategy based on the smallest functional unit of deep models—*parameter clusters*.

Specifically, ignoring non-parameterized layers such as activation layers, Dropout layers, and residual connections, we categorize parameterized layers into two basic types: *aggregation layers* and *processing layers*. Aggregation layers are layers that transform and aggregate information from multiple neurons into new neurons, such as linear layers. Processing layers apply transformations to each neuron individually, such as LayerNorm and BatchNorm.

For **aggregation layers**, take the linear layer as an example. The output $x_i^{(\ell)}$ of the $i$-th neuron in the $\ell$-th linear layer can be expressed as:

$$x_i^{(\ell)} = \sum_{j=1}^{D^{(\ell-1)}} w_{i,j}^{(\ell)} x_j^{(\ell-1)} + b_i^{(\ell)}, \qquad (1)$$

where $x_j^{(\ell-1)}$, $w_{i,j}^{(\ell)}$, and $b_i^{(\ell)}$ are the input, weight parameters, and bias parameters of the linear layer, respectively. $D^{(\ell-1)}$ denotes the input dimension. This shows that the output $x_i^{(\ell)}$ is generated by aggregating transformed information from multiple neurons via parameters $w_i^{(\ell)} \in \mathbb{R}^{D^{(\ell-1)}}$ and $b_i^{(\ell)} \in \mathbb{R}$. Beyond linear layers, this type of transformation-and-aggregation operation is also applicable to convolutional layers.

For **processing layers**, consider the LayerNorm layer.

Given the output $x_i^{(\ell)}$ of the $i$-th neuron in the $\ell$-th linear layer, its LayerNorm operation is:

$$\hat{x}_i^{(\ell)} = \gamma_i^{(\ell)} \left( \frac{x_i^{(\ell)} - \mu_i^{(\ell)}}{\sigma_i^{(\ell)}} \right) + \beta_i^{(\ell)}, \qquad (2)$$

where $x_i^{(\ell)}$ and $\hat{x}_i^{(\ell)}$ are the input and output of the Layer-Norm layer, respectively. $\gamma_i^{(\ell)}$ and $\beta_i^{(\ell)}$ are learnable affine parameters, while $\mu_i^{(\ell)}$ and $\sigma_i^{(\ell)}$ are statistics computed from features. In addition to LayerNorm, similar transformation-and-processing operations are also present in Batch Normalization layers and certain parameterized pooling layers.

Therefore, the smallest functional module corresponds to the sequence of operations that aggregates transformed information from multiple neurons into one neuron (e.g., Eq. (1)) and then applies independent processing to that neuron (e.g., Eq. (2)). Finally, we refer to the parameters within each smallest functional module as a *parameter cluster*, and its manifold is defined as:

$$\mathcal{C}_i^{(\ell)} = \Big( \underbrace{w_{i,1}^{(\ell)}, w_{i,2}^{(\ell)}, \ldots, w_{i,D^{(\ell-1)}}^{(\ell)}, b_i^{(\ell)}}_{\text{Aggregation Parameters}}, \underbrace{\gamma_i^{(\ell)}, \beta_i^{(\ell)}, \ldots}_{\text{Processing Parameters}} \Big),$$

$$(3)$$

where $\mathcal{C}_i^{(\ell)} \in \mathbb{R}^{d_i^{(\ell)}}$ denotes the parameter cluster (or its manifold) of the $i$-th functional unit in layer $\ell$, and $d_i^{(\ell)}$ is its dimensionality.

## 3.2. Implicit Manifold Learning

### 3.2.1. IMPLICIT MANIFOLD AUTO-ENCODER

With the parameter clusters defined above, the full parameter space of a deep model can be viewed as a global manifold in a high-dimensional space, while each parameter-cluster manifold forms a local manifold nested within it. However, deep model parameters contain substantial redundancy (Zhu & Gupta, 2017; Frankle & Carbin, 2018; Deng et al., 2020). Inspired by works that train on low-dimensional parameter manifolds (Li et al., 2022b; Lu et al., 2023), we first learn a more compact low-dimensional manifold $z_i^{(\ell)}$ for each high-dimensional parameter cluster:

$$z_i^{(\ell)} = f_\theta(\mathcal{C}_i^{\uparrow(\ell)}), \mathcal{C}_i^{\uparrow(\ell)} = \mathcal{I}(\mathcal{C}_i^{(\ell)}; d_{\max}), \qquad (4)$$

where $\mathcal{I}$ denotes the interpolation operation used to align the dimensionality of parameter clusters. $\mathcal{C}_i^{\uparrow(\ell)} \in \mathbb{R}^{d_{\max}}$ is the interpolated version of $\mathcal{C}_i^{(\ell)} \in \mathbb{R}^{d_i^{(\ell)}}$, and $d_{\max}$ exceeds all original cluster dimensionalities $d_i^{(\ell)}$. The function $f_\theta$ is an MLP that learns the low-dimensional manifold.

Although the compact low-dimensional manifold removes redundancy, restoring it back to the high-dimensional and inference-ready parameter-cluster manifold remains challenging. We argue that the global manifold in which all

parameter-cluster manifolds are embedded is not just a simple union of clusters but also contains information about their positional relationships. Leveraging this insight, and inspired by Implicit Neural Representations used for complex spatial data (Sitzmann et al., 2019; Li et al., 2022d; Jiang et al., 2020), we propose to implicitly represent the global manifold by reconstructing each parameter cluster using its latent code $z_i^{(\ell)}$ and positional coordinate $(\ell, i)$:

$$\tilde{\mathcal{C}}_i^{(\ell)} = \mathcal{T}(\tilde{\mathcal{C}}_i^{\uparrow(\ell)}; d_i^{(\ell)}), \tilde{\mathcal{C}}_i^{\uparrow(\ell)} = g_\phi(z_i^{(\ell)} \oplus p_i^{(\ell)}), \quad (5)$$

where $\oplus$ denotes concatenation, and $g_\phi$ is an MLP that outputs the reconstructed high-dimensional cluster. $\mathcal{T}$ is a truncation operator that trims the aligned dimensionality $d_{\max}$ back to the original $d_i^{(\ell)}$. Thus, $\tilde{\mathcal{C}}_i^{(\ell)} \in \mathbb{R}^{d_i^{(\ell)}}$ is the truncated reconstruction of $\tilde{\mathcal{C}}_i^{\uparrow(\ell)} \in \mathbb{R}^{d_{\max}}$.

Following prior work (Mildenhall et al., 2021), the coordinate encoding $p_i^{(\ell)}$ is computed using sinusoidal positional encoding to provide Fourier features, allowing $g_\phi$ to better approximate high-frequency variations present in high-dimensional parameter-cluster manifolds:

$$p_i^{(\ell)} = (\mathcal{S}(\ell), \mathcal{S}(i)), \mathcal{S}(v) = (\sin(b^0\pi v), \cos(b^0\pi v), \\ \dots, \sin(b^d\pi v), \cos(b^d\pi v)). \quad (6)$$

This complete framework is named the *Implicit Manifold Auto-Encoder (IMAE)*. In practice, we inject small noise into $z_i^{(\ell)}$ during training to prevent IMAE from overfitting.

### 3.2.2. PARAMETER CLUSTER DIFFERENCE LOSS

The objective of parameter-manifold learning is to minimize the difference between the reconstructed parameter cluster $\tilde{\mathcal{C}}_i^{(\ell)}$ and the original cluster $\mathcal{C}_i^{(\ell)}$. However, the inherently imbalanced numerical distributions of parameter clusters make it challenging to achieve high reconstruction accuracy using standard losses.

Specifically, parameter clusters exhibit both intra-cluster and inter-cluster imbalance. Within a cluster, as shown in Fig. 2(a), most parameter values are concentrated near zero, while large-magnitude values are rare, forming a skewed Gaussian-like distribution. This causes standard regression losses to bias the model toward predicting values near zero. Across clusters, as shown in Fig. 2(a)-(f), clusters from the same layer share similar distributions, but clusters across different layers vary largely in distribution width. Additionally, wide-distribution clusters and narrow-distribution clusters are not evenly represented, causing the model to overfit the majority distribution.

Existing losses for imbalanced regression (Ross & Dollár, 2017; Wang et al., 2019b; Steininger et al., 2021; Yang et al., 2021; Ren et al., 2022) do not apply well to parameter learning, while losses explicitly designed for parameter reconstruction (Schürholt et al., 2022; Wu et al., 2024; Ashkenazi

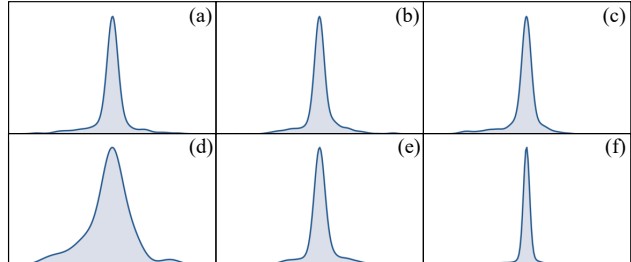

*Figure 2.* Distribution of parameter-cluster values. (a)–(c) show clusters from the same layer, while (d)–(f) show clusters from different layers.

et al., 2022) often lack sufficiently detailed parameter analysis or require complex distillation, making them unsuitable for high-completeness and high-precision parameter modeling. A standard reconstruction loss such as mean squared error (MSE), $\mathcal{L}_{MSE} = \frac{1}{d_i^{(\ell)}} \sum_{d=1}^{d_i^{(\ell)}} (\tilde{\mathcal{C}}_{i,d}^{(\ell)} - \mathcal{C}_{i,d}^{(\ell)})^2$, is easily dominated by imbalanced distributions and tends to predict mean values (Rousseeuw & Leroy, 2003; Ziegel, 2003; Bishop, 2006), making it unsuitable for parameter-cluster reconstruction.

To address this, we extend MSE with two multiplicative weighting terms: an *intra-cluster weighting term* that adjusts the emphasis on small vs. large magnitudes within a cluster, and an *inter-cluster weighting term* that balances clusters with narrow vs. wide distributions. The resulting *Parameter Cluster Difference (PCD) Loss* is:

$$\mathcal{L}_{PCD} = \frac{1}{d_i^{(\ell)}} \sum_{d=1}^{d_i^{(\ell)}} \lambda_{\text{intra}} \lambda_{\text{inter}} (\tilde{\mathcal{C}}_{i,d}^{(\ell)} - \mathcal{C}_{i,d}^{(\ell)})^2,$$

$$\lambda_{\text{intra}} = \exp\left( \alpha \frac{|\mathcal{C}_{i,d}^{(\ell)}| - \min(|\mathcal{C}_i^{(\ell)}|)}{\max(|\mathcal{C}_i^{(\ell)}|) - \min(|\mathcal{C}_i^{(\ell)}|)} \right), \quad (7)$$

$$\lambda_{\text{inter}} = \exp\left( \beta \frac{\max(\hat{\sigma}(\mathcal{C}_i^{(\ell)})) - \sigma(\mathcal{C}_i^{(\ell)})}{\max(\hat{\sigma}(\mathcal{C}_i^{(\ell)})) - \min(\hat{\sigma}(\mathcal{C}_i^{(\ell)}))} \right),$$

where $\lambda_{\text{intra}}$ and $\lambda_{\text{inter}}$ are intra- and inter-cluster weighting factors, respectively. The operators $\min(\cdot)$ and $\max(\cdot)$ compute minimum and maximum values, and $\sigma(\cdot)$ denotes standard deviation. The sequence $\hat{\sigma}(\mathcal{C}_i^{(\ell)})$ contains the standard deviations of *all* parameter clusters in the model, used for normalizing inter-cluster variations. Both weighting terms rely on exponential functions and are controlled by hyperparameters $\alpha$ and $\beta$, which adjust the emphasis placed on magnitude imbalance (intra-cluster) and distribution-width imbalance (inter-cluster).

### 3.3. Implicit Manifold Denoising

Based on the learned low-dimensional parameter-cluster manifold, we introduce a diffusion-model–based denoising

procedure to purify contaminated model parameters. The diffusion model consists of a forward diffusion process and a reverse denoising process.

The goal of the forward process is to gradually add Gaussian noise to the clean low-dimensional parameter-cluster manifold $z_i^{(\ell)}$, so that its distribution approaches a standard Gaussian distribution, forming the prior for reverse denoising. This process is defined as a Markov chain:

$$q(z_{i,(t)}^{(\ell)} \mid z_{i,(t-1)}^{(\ell)}) = \mathcal{N}\left(z_{i,(t)}^{(\ell)}; \sqrt{1 - \beta_{(t)}}\, z_{i,(t-1)}^{(\ell)}, \beta_{(t)}\, \mathbf{I}\right), \tag{8}$$

where $t \in \{1, \dots, T\}$ denotes the diffusion timestep, $z_{i,(0)}^{(\ell)} = z_i^{(\ell)}$ is the clean low-dimensional manifold, $z_{i,(t)}^{(\ell)}$ is the noisy version at step $t$, and $\beta_{(t)} \in (0,1)$ is the noise-schedule parameter.

The reverse process aims to progressively remove noise from the Gaussian prior and recover the clean low-dimensional manifold $z_{i,(0)}^{(\ell)}$. Following variational inference, we learn a parameterized reverse distribution:

$$\begin{aligned} &p_\theta(z_{i,(t-1)}^{(\ell)} \mid z_{i,(t)}^{(\ell)}, \ddot{z}_i^{(\ell)}, p_i^{(\ell)}) \\ &= \mathcal{N}\left(z_{i,(t-1)}^{(\ell)}; \mu_\theta(z_{i,(t)}^{(\ell)}, \ddot{z}_i^{(\ell)}, p_i^{(\ell)}, t), \tilde{\beta}_{(t)}\, \mathbf{I}\right), \end{aligned} \tag{9}$$

where $\ddot{z}_i^{(\ell)}$ denotes the contaminated low-dimensional parameter-cluster manifold, $p_i^{(\ell)}$ is the positional embedding, and $\tilde{\beta}_{(t)} = \frac{1 - \bar{\alpha}_{(t-1)}}{1 - \bar{\alpha}_{(t)}} \beta_{(t)}$ is the variance in the reverse step. In practice, we use a noise-prediction network $h_\psi$ to estimate the noise component. Given the noise component predicted by $h_\psi$, the mean of the reverse transition is computed as:

$$\begin{aligned} &\mu_\theta(z_{i,(t)}^{(\ell)}, \ddot{z}_i^{(\ell)}, p_i^{(\ell)}, t) = \\ &\frac{1}{\sqrt{\alpha_{(t)}}} \left(z_{i,(t)}^{(\ell)} - \frac{\beta_{(t)}}{\sqrt{1 - \bar{\alpha}_{(t)}}} h_\psi(z_{i,(t)}^{(\ell)}, \ddot{z}_i^{(\ell)}, p_i^{(\ell)}, t)\right). \end{aligned} \tag{10}$$

Here, $h_\psi$ predicts the noise component, and $\mu_\theta$ is obtained in closed form from the output of $h_\psi$. In this work, we do not distinguish between the semantic meanings of $\theta$ and $\psi$, and both denote learnable parameters.

During training, the model learns to denoise by minimizing the noise prediction error:

$$L = \mathbb{E}_{z_{i,(0)}^{(\ell)}, p_i^{(\ell)}, t, \epsilon}\left[\|\epsilon - h_\psi(z_{i,(t)}^{(\ell)}, \ddot{z}_i^{(\ell)}, p_i^{(\ell)}, t)\|^2\right]. \tag{11}$$

This loss encourages the model to accurately recover the clean manifold structure conditioned on the contaminated representation $\ddot{z}_i^{(\ell)}$ and the positional embedding $p_i^{(\ell)}$.

During inference, given the contaminated low-dimensional parameter-cluster manifold $\ddot{z}_i^{(\ell)}$ and its positional embedding $p_i^{(\ell)}$, the diffusion model starts by sampling $z_{i,(T)}^{(\ell)}$, and

then iteratively removes noise using:

$$z_{i,(t-1)}^{(\ell)} = \mu_\theta(z_{i,(t)}^{(\ell)}, \ddot{z}_i^{(\ell)}, p_i^{(\ell)}, t) + \sigma_{(t)} z, \quad z \sim \mathcal{N}(0, \mathbf{I}), \tag{12}$$

where $\sigma_{(t)} = \sqrt{\tilde{\beta}_{(t)}}$, and random noise $z$ is added only when $t > 1$ to preserve sample diversity. When $t = 0$, the model outputs $z_{i,(0)}^{(\ell)}$, the purified low-dimensional parameter-cluster manifold. Finally, the decoder $g_\phi$ maps $z_{i,(0)}^{(\ell)}$ back to the high-dimensional parameter space, reconstructing the purified parameter-cluster manifold used for downstream model inference.

## 4. Experiments

We conduct extensive experiments on three representative adverse scenarios that lead to parameter contamination at different granularities, including class imbalance, noisy labels, and backdoor attacks. In addition, a comprehensive set of ablation studies is performed to validate the effectiveness of each component in *Parameter Manifold Purification* (PMP). Due to time and space limitations, the evaluation of PMP on unseen models is conducted within the same scenario. Nevertheless, PMP itself is architecture-agnostic and can be readily applied to multiple model architectures. More detailed experimental settings and additional results are provided in the *Appendix*.

### 4.1. Scenario 1: Imbalanced Samples

#### 4.1.1. SCENARIO DESCRIPTION

In the imbalanced-sample scenario, widely used datasets that induce parameter contamination include CIFAR-10-LT, CIFAR-100-LT, and ImageNet-LT. CIFAR-10-LT and CIFAR-100-LT (Cao et al., 2019) are long-tailed variants of the original CIFAR-10 and CIFAR-100 (Krizhevsky et al., 2009) datasets, constructed by non-uniformly sampling the number of training samples per class to emulate the long-tail distributions commonly observed in real-world data. In this paper, we follow standard practice and set the imbalance factor $r$ to 10, 50, and 100. ImageNet-LT (Liu et al., 2019a) is a long-tailed version of the ImageNet dataset (Russakovsky et al., 2015), obtained by sampling the original dataset using a Pareto distribution with an exponent of 0.6.

#### 4.1.2. COMPARISON WITH SOTA METHODS

We apply PMP to models whose parameters are contaminated by long-tailed data distributions and compare the purified models with state-of-the-art (SOTA) imbalanced learning methods. The competing results are taken from the best performance reported in prior work. As shown by the results, the proposed PMP effectively purifies the parameters of unseen models, leading to performance improvements that surpass existing imbalanced learning approaches.

*Table 1.* Performance of PMP on contaminated parameters of ResNet-50 and ViT-B/16 caused by ImageNet-LT.

| Metric | Head | Medium | Tail | Overall |
|---|---|---|---|---|
| ResNet-50 | | | | |
| cRT (Kang et al., 2019) | 58.8 | 44.0 | 26.1 | 47.3 |
| LWS (Kang et al., 2019) | 57.1 | 45.2 | 29.3 | 47.7 |
| MiSLAS (Zhong et al., 2021) | 62.9 | 50.7 | 34.3 | 52.7 |
| DisAlign (Zhang et al., 2021) | 61.3 | 52.2 | 31.4 | 52.9 |
| BCL (Zhu et al., 2022) | - | - | - | 56.0 |
| NCL (Li et al., 2022a) | - | - | - | 57.4 |
| **PMP (Ours)** | 73.8 | 71.6 | 66.4 | 71.7 |
| ViT-B/16 | | | | |
| LiVT (Xu et al., 2023) | 73.6 | 56.4 | 41.0 | 60.9 |
| BALLAD (Ma et al., 2021) | 79.1 | 74.5 | 69.8 | 75.7 |
| Decoder (Wang et al., 2024b) | - | - | - | 73.2 |
| LIFT w/ TTE (Shi et al., 2024) | 81.3 | 77.4 | 73.4 | 78.3 |
| VL-LTR (Tian et al., 2022) | 84.5 | 74.6 | 59.3 | 77.2 |
| GML (Suh & Seo, 2023) | - | - | - | 78.0 |
| **PMP (Ours)** | 85.1 | 84.1 | 80.2 | 82.6 |

*Table 2.* Performance of PMP on contaminated parameters of ResNet-34 caused by synthetic noisy labels in CIFAR-10 and CIFAR-100.

| Dataset | CIFAR-10 | | CIFAR-100 | |
|---|---|---|---|---|
| Noise Ratio | 20% | 80% | 20% | 80% |
| CE | 87.78 | 67.43 | 68.35 | 27.91 |
| Forward (Patrini et al., 2017) | 87.99 | 54.64 | 39.77 | 8.99 |
| GCE (Zhang & Sabuncu, 2018) | 89.83 | 64.07 | 61.77 | 25.91 |
| SL (Wang et al., 2019c) | 89.83 | 68.12 | 62.81 | 29.78 |
| ELR (Liu et al., 2020a) | 91.16 | 52.48 | 74.21 | 20.23 |
| SOP (Liu et al., 2022) | 93.18 | 68.32 | 74.67 | 30.20 |
| **PMP(Ours)** | 96.40 | 94.20 | 79.10 | 69.14 |

For example, as shown in Table 1, on ImageNet-LT with ViT-B/16, PMP achieves an overall performance of 82.6%, surpassing the previous SOTA result of 78.0% by 4.6%.

### 4.2. Scenario 2: Imprecise Labels

#### 4.2.1. SCENARIO DESCRIPTION

In the scenario of imprecise labels, parameter contamination primarily arises from synthetic noisy labels and real-world noisy labels. For synthetic noise, following established settings (Tanaka et al., 2018), we generate both symmetric and asymmetric label noise based on CIFAR-10 and CIFAR-100. In this work, the symmetric noise rates $r$ are set to the commonly used values of 20% and 80%, while the asymmetric noise rate $r$ is set to 40%. For real-world noisy labels, we adopt the recently introduced CIFAR-N benchmark (Wei et al., 2021b), which includes CIFAR-10-N and CIFAR-100-N with labels collected via Amazon Mechanical Turk, thereby reflecting realistic human-annotated noise patterns.

#### 4.2.2. COMPARISON WITH SOTA METHODS

As shown in Tab. 2, we conduct PMP on models contaminated by a variety of noisy-label settings and compare the purified models with conventional SOTA noisy-label learning methods, whose reported performance is taken from previously published work. The results demonstrate that PMP effectively addresses parameter contamination caused by both synthetic and real noisy labels, and the purified parameters consistently outperform those produced by existing approaches. For example, in Table 2, for CIFAR-10 with 80% symmetric noise using ResNet-34, PMP improves the model accuracy to 94.20%, surpassing the performance of current SOTA methods.

### 4.3. Scenario 3: Backdoor Attacks

#### 4.3.1. SCENARIO DESCRIPTION

In the backdoor attack scenario, we evaluate parameter purification using two widely adopted datasets: CIFAR-10 (Krizhevsky et al., 2009) and Tiny-ImageNet (Le & Yang, 2015). Following prior studies (Zhu et al., 2024), we consider eight representative SOTA backdoor attack methods, including BadNets (BadNets-A2O and BadNets-A2A) (Gu et al., 2019), Blended Attack (Chen et al., 2017), Input-Aware Dynamic Backdoor Attack (Nguyen & Tran, 2020), Low-Frequency (LF) Attack (Zeng et al., 2021b), Sample-Specific Backdoor Attack (SSBA) (Li et al., 2021c), Trojan Attack (Liu et al., 2018b), and WaNet (Nguyen & Tran, 2021). To ensure fair comparison across methods, we follow the default configurations provided in BackdoorBench (Wu et al., 2022), including trigger patterns, trigger sizes, and target labels.

#### 4.3.2. COMPARISON WITH SOTA METHODS

We conduct PMP on models whose parameters are contaminated by a diverse set of backdoor attacks, and compare the purified models with conventional SOTA backdoor defense methods, using the best results reported in prior work. The results consistently show that PMP effectively mitigates backdoor-induced parameter contamination and restores model behavior. For example, as shown in Table 3, for PreAct-ResNet-18 under the BadNets-A2O attack on Tiny-ImageNet, applying PMP increases the clean accuracy to 56.15% while reducing the attack success rate to 0.21%, outperforming existing SOTA backdoor defense approaches.

### 4.4. Ablation Study

#### 4.4.1. LOCAL MANIFOLD POSITIONAL INFORMATION

As shown in Figure 3(a), we conduct an ablation study on the local manifold positional information $p_i^{(\ell)}$ in Equation (5) to evaluate its contribution to IMAE. The results demonstrate that incorporating $p_i^{(\ell)}$ substantially enhances the accuracy of parameter reconstruction. Specifically, with-

*Table 3.* Performance of PMP on contaminated parameters of ResNet-18 caused by backdoor attacks in Tiny-ImageNet.

| Attack | BadNets-A2O | | BadNets-A2A | | Blended | | Input-Aware | | LF | | SSBA | | Trojan | | WaNet | |
|---|---|---|---|---|---|---|---|---|---|---|---|---|---|---|---|---|
| Metric | ACC | ASR | ACC | ASR | ACC | ASR | ACC | ASR | ACC | ASR | ACC | ASR | ACC | ASR | ACC | ASR |
| Backdoored | 56.05 | 99.96 | 55.71 | 27.3 | 55.95 | 99.60 | 57.42 | 99.70 | 55.13 | 98.60 | 55.35 | 97.70 | 56.14 | 99.97 | 58.34 | 99.99 |
| FP (Liu et al., 2018a) | 48.81 | 0.66 | 47.88 | 3.19 | 50.58 | 57.89 | 52.38 | 0.13 | 48.18 | 63.83 | 48.06 | 52.25 | 45.96 | 8.88 | 50.35 | 1.37 |
| NAD (Li et al., 2021d) | 48.35 | 0.27 | 48.29 | 2.30 | 55.22 | 98.88 | 57.42 | 0.07 | 49.61 | 58.01 | 47.67 | 69.47 | 48.83 | 1.01 | 50.02 | 0.87 |
| NC (Wang et al., 2019a) | 56.12 | 99.90 | 54.12 | 18.72 | 54.50 | 96.07 | 53.46 | 2.48 | 53.08 | 90.48 | 53.30 | 0.26 | 54.43 | 1.54 | 57.81 | 96.50 |
| ANP (Wu & Wang, 2021) | 47.34 | 0.00 | 40.70 | 2.39 | 43.21 | 43.80 | 50.56 | 0.00 | 41.75 | 65.98 | 41.83 | 14.24 | 45.36 | 0.53 | 30.34 | 0.00 |
| i-BAU (Zeng et al., 2021a) | 51.63 | 95.92 | 53.52 | 12.89 | 50.76 | 95.58 | 55.49 | 0.46 | 53.65 | 94.27 | 52.39 | 84.64 | 51.85 | 99.15 | 53.04 | 69.82 |
| EP (Zheng et al., 2022) | 54.00 | 0.02 | 54.79 | 1.28 | 56.32 | 88.88 | 57.33 | 0.03 | 54.86 | 93.20 | 55.56 | 66.67 | 54.47 | 0.12 | 57.06 | 0.20 |
| NPD (Zhu et al., 2024) | 49.79 | 2.51 | 49.94 | 5.57 | 49.62 | 0.12 | 53.75 | 5.93 | 49.94 | 2.48 | 49.25 | 0.01 | 49.43 | 0.51 | 52.64 | 0.24 |
| **PMP(Ours)** | 56.15 | 0.21 | 55.28 | 1.85 | 54.42 | 0.11 | 55.84 | 0.28 | 55.12 | 0.63 | 55.09 | 0.12 | 55.23 | 0.31 | 56.16 | 0.35 |

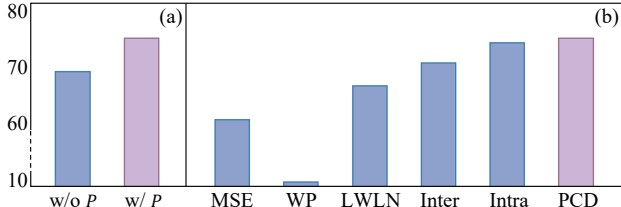

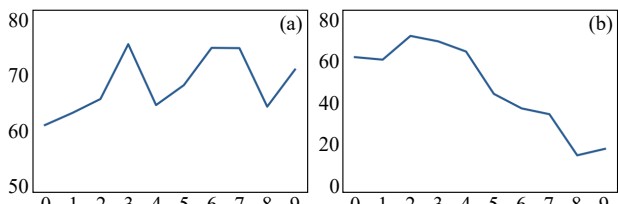

*Figure 3.* Impact of local manifold position information (subplot a) and parameter cluster discrepancy loss (subplot b) on the accuracy of parameter manifold learning. "w/ $P$" and "w/o $P$" represent the cases with and without the use of position information, respectively. "WP"(Wu et al., 2024) and "LWLN"(Schürholt et al., 2022) are loss functions specifically designed for parameter learning. "Inter", "Intra", and "PCD" refer to parameter discrepancy losses with only intra-cluster weighting, only inter-cluster weighting, and both intra- and inter-cluster weighting, respectively.

*Figure 4.* Effect of hyperparameters $\alpha$ (subfigure a) and $\beta$ (subfigure b) in the parameter cluster difference loss on the accuracy of implicit manifold learning. The x-axis represents the values of $\alpha$ and $\beta$, while the y-axis represents the corresponding accuracy of the reconstructed parameters.

out local manifold positional information, the reconstructed parameters achieve an accuracy of only $69.64$. In contrast, when $p_i^{(\ell)}$ is included, the reconstruction accuracy increases to $75.32$, highlighting its importance for accurately modeling local parameter-cluster manifolds nested within the global high-dimensional manifold.

### 4.4.2. PARAMETER CLUSTER DISCREPANCY LOSS

As shown in Figure 3(b), we conduct ablation experiments on the weighting terms $\lambda_{\text{intra}}$ and $\lambda_{\text{inter}}$ in Equation (7) to assess the effectiveness of the proposed parameter cluster discrepancy (PCD) loss. We further compare it with several representative loss functions designed for parameter learning. The results clearly show that the PCD loss provides the most effective restoration of the local parameter-cluster manifolds. Specifically, using the conventional MSE loss yields a reconstruction accuracy of $61.42$. WP Loss and LWLN achieve accuracies of $10.86$ and $67.25$, respectively. In contrast, the complete weighted PCD loss attains an accuracy of $75.32$, demonstrating its superior ability to model and recover parameter-cluster manifolds.

### 4.4.3. HYPERPARAMETERS $\alpha$ AND $\beta$ IN PCD LOSS

As shown in Fig. 4, we analyze the impact of the hyperparameter $\alpha$ in the intra-cluster weighting term $\lambda_{\text{intra}}$ and the

hyperparameter $\beta$ in the inter-cluster weighting term $\lambda_{\text{inter}}$ (from Eq. 7) on the accuracy of implicit manifold learning. From Fig. 4(a), we observe that the reconstruction accuracy is relatively sensitive to the choice of $\alpha$, achieving its peak when $\alpha = 3$. In comparison, Fig. 4(b) shows that the reconstruction accuracy gradually decreases as $\beta$ increases, with the highest accuracy obtained when $\beta = 2$. This behavior can be attributed to the fact that the intra-cluster weighting term is more strongly affected by outliers within parameter clusters than the inter-cluster weighting term, making its choice of hyperparameters more critical for accurate manifold modeling.

## 5. Conclusion

Inspired by image denoising, this paper introduces a novel task, *parameter purification*, which serves as a general optimization paradigm capable of directly improving performance-degraded models. Furthermore, we propose the first method for parameter purification, termed *parameter manifold purification* (PMP). Extensive experiments across multiple adverse scenarios that lead to parameter contamination demonstrate that PMP can effectively purify contaminated parameters and significantly outperform conventional model optimization approaches. Beyond the scenarios evaluated in this work, the general optimization paradigm of parameter purification holds strong potential for mitigating performance degradation caused by model quantization, adversarial attacks, and other challenging con-

ditions. We hope this work inspires further exploration of parameter purification as a universal optimization paradigm and advances the development of efficient techniques for robust and high-performance artificial intelligence systems.

## Acknowledgements

This work is supported by National Natural Science Foundation of China (62376248), the Starry Night Science Fund of Zhejiang University Shanghai Institute for Advanced Study (Grant No. SN-ZJU-SIAS-001), and Information Technology Center and State Key Lab of CAD&CG ,Zhe-Jiang University.

## Impact Statement

This paper introduces Parameter Manifold Purification, which attributes model performance degradation to the contamination of model parameters. Repairing parameter contamination caused by backdoor attacks can reduce attack success rates and restore benign model behavior, thereby improving the security and trustworthiness of deployed models.

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

# Appendix

In addition to the code provided at `https://parameter-generation.github.io/`, this appendix includes the following contents:

## A. More Related Work

### A.1. Parameter Properties Prediction

Predicting parameter properties is another line of work closely related to ours, as it also involves learning representations of neural network parameters. Parameter properties prediction aims to infer model-related attributes such as accuracy or generalization ability from the model parameters (Martin et al., 2021; Baker et al., 2017; Yak et al., 2019; Jiang et al., 2018; Corneanu et al., 2020; Martin & Mahoney, 2017; 2021). One representative direction is to use an additional neural network to model the parameters of the target network for the purpose of property prediction (Schürholt et al., 2021). Existing studies in this direction can be categorized into representation-based (Schürholt et al., 2021; 2022; 2024), mapping-based (Eilertsen et al., 2020; Unterthiner et al., 2020; Herrmann et al., 2024; Langosco et al., 2023; Andreis et al.), symmetry-based (Navon et al., 2023; Zhou et al., 2024a;b; Tran et al., 2024b;a), graph-based (Kofinas et al., 2024; Lim et al., 2023), and probe-based (Kahana et al., 2024; Horwitz et al., 2025) approaches.

For example, representation-based methods typically employ an auto-encoder to learn a hyper-representation of network parameters, which is then used for downstream performance prediction (Schürholt et al., 2021). Mapping-based methods directly input model parameters into another neural network to establish a mapping between parameters and model attributes (Unterthiner et al., 2020). Moreover, symmetry-based approaches exploit permutation symmetries in deep neural networks (Navon et al., 2023), graph-based methods represent deep models as computational graphs of parameters (Kofinas et al., 2024), and probe-based methods learn a probe network to characterize the target network and predict its parameter properties (Kahana et al., 2024).

Although these approaches also utilize neural networks to process parameters, they fundamentally differ from ours. These methods focus solely on predicting the properties of parameters. In contrast, we perform manifold learning and purification on contaminated parameters to obtain cleaner and more accurate parameters.

### A.2. Implicit Neural Representation

Another category of related work is Implicit Neural Representation, which has proven to be a powerful tool for representing various signals using multilayer perceptrons that map coordinates to signal values. This framework has been widely applied to 3D scenes (Sitzmann et al., 2019; Li et al., 2022d; Jiang et al., 2020; Chabra et al., 2020; Niemeyer et al., 2020; Oechsle et al., 2019; Mildenhall et al., 2021), images (Chen et al., 2021b; Sitzmann et al., 2020; Dupont et al., 2021), videos (Chen et al., 2021a; 2022; He et al., 2023), and pose estimation (Yen-Chen et al., 2021). Building on the expressive power of implicit neural representations, several studies have explored learning the parameters of INRs using deep learning, similar to hypernetworks (De Luigi et al., 2023; Dupont et al., 2022; Erkoç et al., 2023; Shamsian et al., 2024), for downstream tasks such as signal synthesis (Erkoç et al., 2023) or signal classification (De Luigi et al., 2023). Recently, some works have

*Table 4.* Comparison between PMP and representative weight-space diffusion methods. PMP differs fundamentally by addressing the new problem of **parameter purification**, introducing a functional-unit–aware parameter clustering and a high-precision latent denoising mechanism.

| Dimension | PMP (Ours) | p-diff | D2NWG | HyperDiffusion |
|---|---|---|---|---|
| Core objective | **Parameter purification**: recovering clean weights from contaminated models | Parameter generation | Task-conditioned weight generation | Implicit field weight generation |
| Problem formulation | **Unified view** of long-tailed, noisy-label and backdoor settings as "parameter contamination" | Generate new performant models | Generate transferable weights for new tasks | Generate MLP weights for implicit fields |
| Weight structure modeling | **Parameter clusters** (functional-unit level) + implicit manifold autoencoder | Encoded partial parameter subsets | Latent representation of full models | No explicit structural partition, whole-MLP representation |
| Diffusion space | **Low-dimensional latent space of parameter clusters** (conditional) | Latent diffusion (unconditional) | Conditional latent diffusion | High-dimensional weight-space diffusion |
| Optimization focus | **High-precision restoration** via PCD loss (handling intra-/inter-cluster imbalance) | Reconstruction + generation | Distribution modeling + generation | High-dimensional generative modeling |
| Need for target data | **No**: inference is data-free, only uses the contaminated model | No | Requires task/dataset embedding | No |
| Main application scenarios | **Unified post-hoc repair**: long-tailed, noisy-label, and multiple backdoor attacks | Model diversity and generation | Multi-task transfer and initialization | Shape/scene implicit field generation |
| Model completeness | **Full-network coverage** with functional-unit consistency | Partial selected parameters | Full models, but without functional-unit reasoning | Whole MLP networks |

proposed modeling neural network parameters directly using INRs.

For instance, Ashkenazi et al. (2022) employed INRs to model the weights of pretrained convolutional networks. However, their representation can only model convolutional weights and requires auxiliary knowledge distillation to enhance performance. Yang & Wang (2025) proposed modeling convolutional network parameters with INRs for network resizing, but their approach requires multiple INRs to model different network partitions and heavily depends on parameter smoothness.

Unlike the above approaches, our work draws inspiration from implicit neural representations and proposes an implicit manifold auto-encoder that maps parameters within each cluster to a low-dimensional manifold. Coordinate embeddings indicate the position of the low-dimensional manifold within the global parameter manifold, enabling precise parameter modeling. The learned low-dimensional manifolds further serve as the basis for manifold purification to improve model performance.

## B. Differences from Existing Work

Although our parameter manifold purification method also incorporates diffusion models (Ho et al., 2020; Nichol & Dhariwal, 2021; Croitoru et al., 2023), similar to recent parameter diffusion approaches (Xie et al.; Rangel et al., 2024; Du et al., 2023; Zhang et al., 2024; Wang et al., 2024a; Wei et al., 2024; Soro et al., 2024; Nava et al., 2022), our work differs from them in four fundamental aspects.

- First, the target task is different. We focus on parameter purification, aiming to establish a general optimization paradigm. Unlike methods that only generate usable parameters, parameter manifold purification must progressively purify contaminated parameters to further enhance model performance, which imposes a higher requirement on parameter precision.

- Second, the parameter learning structure differs. Through a detailed analysis of neural network parameter characteristics, we introduce a novel parameter-cluster manifold learning structure, providing better interpretability and scalability.

- Third, the parameter learning methodology differs. To achieve high-precision parameter modeling, we propose an implicit manifold auto-encoder and a parameter-cluster discrepancy loss to accurately learn low-dimensional manifolds. We further incorporate a conditional diffusion model for manifold denoising.

- Fourth, the coverage of parameters differs. Unlike most parameter generation methods that only support small networks or a limited number of parameters, our parameter manifold purification method can learn and purify the entire parameter space of large mainstream neural networks.

To highlight the novelty of our method, we also compare PMP with representative weight-space diffusion approaches, including p-diff (Wei et al., 2024), D2NWG (Soro et al., 2024) and HyperDiffusion (Erkoç et al., 2023), as shown in Table 4. Unlike prior methods that focus mainly on *parameter generation*, *task-conditioned weight synthesis*, or *implicit field generation*, PMP is the first to formulate and solve a unified **parameter purification** problem: treating long-tailed imbalance, noisy labels, and backdoor attacks as forms of *parameter contamination* and performing data-free model repair. Moreover, PMP introduces a functional-unit–aware parameter cluster formulation, an implicit manifold autoencoder and a PCD loss tailored for high-precision parameter restoration, which are absent in existing diffusion-based weight-space methods.

Existing methods cannot be directly applied to the parameter purification task by simply replacing the loss function; instead, they require substantial modifications to their pipelines, including how to process full-network parameter clusters and how to design conditional inputs. In the subsequent sections of the appendix, we compare the proposed method with several representative diffusion-based weight-generation approaches, evaluating their performance specifically under the parameter purification setting.

## C. More Details on Parameter Cluster Partitioning

In addition to linear layers, convolutional layers are also typical aggregation layers. The transformation-and-aggregation operation described in the main text is equally applicable to convolutional layers. The key difference lies in the parameterization: in linear layers, the weight vector $w_i^{(\ell)} \in \mathbb{R}^{D^{(\ell-1)}}$ consists of $D^{(\ell-1)}$ scalar values, whereas in convolutional layers, $w_i^{(\ell)} \in \mathbb{R}^{D^{(\ell-1)} \times h^{(\ell)} \times w^{(\ell)}}$ is composed of $D^{(\ell-1)}$ two-dimensional convolutional kernels, where $h^{(\ell)}$ and $w^{(\ell)}$ denote the height and width of the convolutional kernel at layer $\ell$, respectively.

Besides LayerNorm, Batch Normalization is another representative processing layer. The transformation-and-processing operation discussed in the main text also applies to BatchNorm layers. The distinction in parameters lies in the statistical terms: for LayerNorm, the statistics $\mu_i^{(\ell)}$ and $\sigma_i^{(\ell)}$ are computed on-the-fly from each individual sample, whereas for BatchNorm, $\mu_i^{(\ell)}$ and $\sigma_i^{(\ell)}$ are computed over the entire batch during training and remain fixed at test time. Beyond normalization operations, certain parameterized pooling layers (Gulcehre et al., 2014) also perform transformation-and-processing operations and are therefore categorized as processing layers.

Moreover, for more complex model components, such as the canonical self-attention module in Transformers, the core $QKV$ operations can be decomposed into a composition of multiple linear transformations. Consequently, these operations can also be regarded as aggregation layers, and the corresponding parameters follow the same parameter cluster partitioning rules as aggregation layers. In contrast, residual connections commonly used in Transformers do not involve learnable parameters and thus do not require parameter cluster partitioning.

In summary, any module in modern deep learning architectures can be progressively decomposed into two types of parameterized layers: aggregation layers and processing layers. Defining a minimal functional module as the composition of aggregation and processing operations is consistent with prior interpretability studies (Hu et al., 2024b). Aggregation operations are responsible for forming distinct semantic representations, while subsequent processing operations further refine these semantics. As a result, the parameters within a minimal functional module consistently maintain the same semantic information. Depending on the specific model architecture, each parameter cluster derived from such minimal functional modules typically contains either zero or one aggregation operation, along with zero or more processing operations. This parameter cluster partitioning strategy is more interpretable and facilitates subsequent parameter manifold purification.

*Table 5.* Number and dimensionality of parameter clusters for several model architectures.

| Architecture | Number of Clusters | Cluster Dim. |
|---|---|---|
| PreAct-ResNet-18 | 4810 | 4610 |
| ResNet-34 | 8522 | 4610 |
| ResNet-50 | 27560 | 4610 |
| ViT-B/16 | 84712 | 3075 |

## D. More Details of Implicit Manifold Denoising

For the Markov chain, the sample at step $t$ can be directly obtained from $z_{i,(0)}^{(\ell)}$ using a closed-form expression:

$$q(z_{i,(t)}^{(\ell)} \mid z_{i,(0)}^{(\ell)}) = \mathcal{N}\left(z_{i,(t)}^{(\ell)}; \sqrt{\bar{\alpha}_{(t)}}\, z_{i,(0)}^{(\ell)}, (1 - \bar{\alpha}_{(t)})\,\mathbf{I}\right), \tag{13}$$

where $\alpha_{(t)} = 1 - \beta_{(t)}$ and $\bar{\alpha}_{(t)} = \prod_{s=1}^{t} \alpha_{(s)}$. As $t \to T$, we have $\bar{\alpha}_{(t)} \to 0$, and thus the distribution of $z_{i,(T)}^{(\ell)}$ approaches the standard Gaussian distribution $\mathcal{N}(0, \mathbf{I})$.

It is important to note that although sampling begins with $z_{i,(T)}^{(\ell)} \sim \mathcal{N}(0, \mathbf{I})$, the contaminated representation $\ddot{z}_i^{(\ell)}$ and the positional embedding $p_i^{(\ell)}$ are incorporated at every time step through the denoising network $h_\psi(z_{i,(t)}^{(\ell)}, \ddot{z}_i^{(\ell)}, p_i^{(\ell)}, t)$ within $\mu_\theta(\cdot)$. Consequently, contamination information is continuously injected and constrained along the entire reverse chain, yielding samples from the conditional posterior $p_\theta(z_{i,(0)}^{(\ell)} \mid \ddot{z}_i^{(\ell)}, p_i^{(\ell)})$.

## E. Additional Experimental Settings for Parameter Manifold Purification

### E.1. Training Data for PMP

Training parameter manifold purification requires a large number of pairs of contaminated parameters and clean parameters. For contaminated parameters, we train models under the three adverse scenarios discussed in the main text, including long-tailed distributions, noisy labels, and backdoor attacks, using each scenario's default training datasets and training configurations, resulting in a variety of contaminated parameters. For clean parameters, we train higher-performing models using the corresponding original clean datasets and settings for each scenario, thereby obtaining a large number of clean parameters.

These contaminated and clean parameters are split into training and testing sets at an 8:2 ratio, where the training set is used to train parameter manifold purification, and the testing set is used to evaluate purification performance. Regarding the model architectures from which contaminated and clean parameters are collected, we use mainstream convolutional neural networks and Transformer architectures, both containing large numbers of parameters. Moreover, a single PMP model can simultaneously process multiple architectures within the same scenario, and PMP is evaluated on unseen new models.

Because PMP operates at the parameter-cluster level, we provide, as a reference for training sample quantity, the number and dimensionality of parameter clusters from individual models of several architectures, as shown in Table 5. Each architecture contains a very large number of parameter clusters, which provides abundant data to support PMP training. Meanwhile, the dimensionality of each parameter cluster is relatively small (<5000), substantially reducing per-iteration computational cost during training and inference. As more models participate in training, the total number of parameter clusters increases proportionally, offering a robust data foundation for PMP fitting.

Details about the datasets, training configurations, and model architectures used to generate contaminated and clean parameters for each scenario can be found in the subsequent scenario-specific sections of the appendix.

### E.2. Network Architectures for PMP

For the proposed Implicit Manifold Auto-Encoder (IMAE), both the encoder and decoder consist of four linear layers. The encoder input dimensionality and decoder output dimensionality match the maximum parameter-cluster dimension (typically below 5000). Each parameter cluster is projected into a 128-dimensional low-dimensional manifold. As a result, the IMAE architecture is compact and easy to train.

*Table 6.* Performance of PMP on contaminated parameters of ResNet-32 caused by CIFAR-10-LT and CIFAR-100-LT.

| Dataset | CIFAR-10-LT | | | CIFAR-100-LT | | |
|---|---|---|---|---|---|---|
| Factor | 100 | 50 | 10 | 100 | 50 | 10 |
| ResNet-32 | | | | | | |
| Focal Loss (Ross & Dollár, 2017) | 70.4 | 76.7 | 86.7 | 38.4 | 44.3 | 55.8 |
| CB-Focal (Cui et al., 2019) | 74.6 | 79.3 | 87.5 | 39.6 | 45.2 | 58.0 |
| LDAM-DRW (Cao et al., 2019) | 77.0 | 81.0 | 88.2 | 42.0 | 46.6 | 58.7 |
| BBN (Zhou et al., 2020) | 79.8 | 81.2 | 88.3 | 42.6 | 47.0 | 59.1 |
| SSP (Yang & Xu, 2020) | 77.8 | 82.1 | 88.5 | 43.4 | 47.1 | 58.9 |
| TSC (Li et al., 2022a) | 79.7 | 82.9 | 88.7 | 43.8 | 47.4 | 59.0 |
| Causal Model (Tang et al., 2020) | 80.6 | 83.6 | 88.5 | 44.1 | 50.3 | 59.6 |
| Hybrid-SC (Wang et al., 2021) | 81.4 | 85.4 | 91.1 | 46.7 | 51.9 | 63.1 |
| MSA-LDAM (Li et al., 2021a) | 80.7 | 84.3 | 89.7 | 48.0 | 52.3 | 61.3 |
| ResLT (Cui et al., 2022) | 82.4 | 85.2 | 89.7 | 48.2 | 52.7 | 62.0 |
| LA (Menon et al., 2020) | 84.3 | 87.1 | 90.9 | 50.5 | 54.9 | 64.0 |
| BCL (Zhu et al., 2022) | 84.5 | 87.2 | 91.1 | 51.9 | 56.4 | 64.6 |
| ProCo (Xiao et al., 2022) | 85.9 | 88.2 | 91.9 | 52.8 | 57.1 | 65.5 |
| KCL (Kang et al., 2020) | 77.6 | 81.7 | 88.0 | 42.8 | 46.3 | 57.6 |
| Remix (Chou et al., 2020) | 75.4 | - | 88.2 | 41.9 | 49.5 | 59.4 |
| UniMix (Xu et al., 2021) | 82.8 | 84.3 | 89.7 | 45.5 | 51.1 | 61.3 |
| SMC (Jeong & Kim, 2022) | - | - | - | 48.9 | 52.3 | 62.5 |
| **PMP(Ours)** | **89.9** | **91.2** | **93.1** | **60.5** | **65.6** | **68.2** |

*Table 7.* Further performance of PMP on contaminated parameters of ResNet-34 caused by synthetic noisy labels in CIFAR-10 and CIFAR-100.

| Dataset | CIFAR-10 | | | | CIFAR-100 | | | |
|---|---|---|---|---|---|---|---|---|
| Noise Radio | 20% | 40% | 60% | 80% | 20% | 40% | 60% | 80% |
| CE | 87.78 | 82.65 | 75.54 | 67.43 | 68.35 | 54.17 | 46.98 | 27.91 |
| Forward (Patrini et al., 2017) | 87.99 | 83.25 | 74.96 | 54.64 | 39.77 | 31.05 | 19.12 | 8.99 |
| GCE (Zhang & Sabuncu, 2018) | 89.83 | 84.13 | 82.61 | 64.07 | 61.77 | 62.27 | 54.82 | 25.91 |
| SL (Wang et al., 2019c) | 89.83 | 87.13 | 82.81 | 68.12 | 62.81 | 68.28 | 59.28 | 29.78 |
| ELR (Liu et al., 2020a) | 91.16 | 86.12 | 73.86 | 52.48 | 74.21 | 60.26 | 59.28 | 20.23 |
| SOP (Liu et al., 2022) | 93.18 | 90.09 | 86.76 | 68.32 | 74.67 | 70.12 | 60.26 | 30.20 |
| **PMP(Ours)** | **96.42** | **93.22** | **93.51** | **94.23** | **79.14** | **74.64** | **70.11** | **69.14** |

For the diffusion model used for manifold denoising, we adopt the standard DDPM. The U-Net used to predict noise is primarily composed of one-dimensional convolutional layers with residual connections. Its input consists of the cluster manifold and cluster coordinates, and its output is the predicted parameter-cluster noise at a given timestep.

Although the overall architecture of parameter manifold purification is simple, it is highly effective. This simplicity also leaves room for future customization and enhancement of PMP performance.

### E.3. Training Settings for PMP

When training the IMAE, the hyperparameter $\alpha$ in the parameter-cluster discrepancy loss is set to 1 for convolutional neural network architectures and 6 for Transformer architectures (reflecting differences in value distributions across parameter clusters induced by architectural properties). The hyperparameter $\beta$ is fixed at 1 for all architectures, and the hyperparameter $b$ used in cluster coordinates is set to 2.

We use the AdamW optimizer with a weight decay of $1 \times 10^{-4}$, train for $1 \times 10^4$ iterations, and use an initial learning rate of $1 \times 10^{-4}$. A cosine annealing learning rate schedule with $T_{\max} = 1 \times 10^4$ is applied.

When training the diffusion model, we use the AdamW optimizer with a weight decay of $1 \times 10^{-4}$, train for $1 \times 10^3$ iterations, and use an initial learning rate of $1 \times 10^{-4}$. A two-stage learning rate decay strategy with a 1:1:1 ratio is applied, where the learning rate is reduced by a factor of 10 at each stage until reaching $1 \times 10^{-6}$.

*Table 8.* Further performance of PMP on contaminated parameters of ResNet-34 caused by synthetic noisy labels in CIFAR-10 and CIFAR-100.

| Dataset | CIFAR-10 | | | | CIFAR-100 | | | |
|---|---|---|---|---|---|---|---|---|
| | Sym. | | | Asym. | Sym. | | | Asym. |
| Noise Radio | 20% | 50% | 80% | 40% | 20% | 50% | 80% | 40% |
| CE | 87.8 | 68.2 | 67.7 | 79.4 | 68.4 | 44.0 | 27.9 | 67.3 |
| MixUp (Zhang, 2017) | 93.5 | 87.9 | 72.3 | - | 69.9 | 57.3 | 33.6 | - |
| DivideMix (Li et al., 2020) | 96.1 | 94.6 | 93.4 | 93.4 | 77.1 | 74.6 | 60.8 | 72.1 |
| ELR+ (Liu et al., 2020a) | 95.8 | 94.8 | 93.3 | 93.0 | 77.7 | 74.8 | 60.8 | 77.5 |
| SOP+ (Liu et al., 2022) | 96.3 | 95.5 | 94.0 | 93.8 | 78.8 | 75.9 | 63.3 | 78.0 |
| **PMP(Ours)** | 96.4 | 95.8 | 94.2 | 94.1 | 79.1 | 76.4 | 69.1 | 78.3 |

# F. Experimental Setup for Scenario 1 (Imbalanced Samples)

## F.1. Networks and Parameters

To fully validate the effectiveness of the proposed PMP, we use commonly adopted network architectures in long-tail distribution scenarios to obtain both contaminated and purified parameters for the experiments. Specifically, for constructing contaminated parameters on the CIFAR-10/100-LT datasets, we follow (Zhu et al., 2022) and use ResNet-32 (He et al., 2016a) as the backbone network. For ImageNet-LT, we follow (Zhu et al., 2022) and prior works (Shi et al., 2024) and use ResNet-50 (He et al., 2016a) and ViT-B/16 (Dosovitskiy, 2020) as backbone networks. During training, we employ AutoAug (Cubuk et al., 2019) and Cutout (DeVries, 2017) as data augmentation strategies. We use the Adam optimizer with weight decay of $5 \times 10^{-4}$, an initial learning rate of $1 \times 10^{-3}$ with cosine annealing for learning rate decay, a batch size of 256, and train for 300 epochs, followed by multiple rounds of fine-tuning to obtain more contaminated parameters. For the clean parameter data, we train on the original CIFAR-10/100 and ImageNet datasets using the corresponding backbone networks and training settings, again fine-tuning for several rounds to obtain additional clean parameters. Notably, when obtaining clean parameters for ImageNet, we use the pre-trained models provided by TorchVision for fine-tuning.

## F.2. Metrics

In addition to reporting overall Top-1 accuracy, we follow the evaluation protocol introduced by (Liu et al., 2019a) to report accuracy for three categories: head classes (with more than 100 images), medium classes (with 20 to 100 images), and tail classes (with fewer than 20 images).

# G. Experimental Setup for Scenario 2 (Imprecise Labels)

## G.1. Networks and Parameters

To thoroughly validate the effectiveness of the proposed PMP, we employ commonly used network architectures for the noisy label to obtain both contaminated and clean parameters for the experiments. Specifically, for generating contaminated parameters on CIFAR-10 and CIFAR-100 noisy label datasets, we follow prior works (Xiao et al., 2022) and use ResNet-18 and ResNet-34 as backbone networks. During training, we use AutoAug (Cubuk et al., 2019) and Cutout (DeVries, 2017) as data augmentation strategies. We employ the Adam optimizer with a weight decay of $5 \times 10^{-4}$, an initial learning rate of $1 \times 10^{-3}$ with a cosine annealing learning rate decay, a batch size of 128, and train for 300 epochs, followed by several rounds of fine-tuning to obtain additional contaminated parameters. For clean parameters, we train on the original CIFAR-10 and CIFAR-100 datasets using the corresponding backbone networks and training settings, also fine-tuning for multiple rounds to obtain more clean parameters.

## G.2. Metrics

For model evaluation, we primarily use Top-1 accuracy to reflect the performance of the model.

*Table 9.* Performance of PMP on contaminated parameters of ResNet-18 caused by real noisy labels in CIFAR-10-N and CIFAR-100-N.

| Dataset | CIFAR-10-N | | | | | CIFAR-100-N |
|---|---|---|---|---|---|---|
| Noise Type | Aggregate | Random1 | Random 2 | Random 3 | Worst | Noisy-Fine |
| CE | 92.58 | 88.3 | 87.46 | 88.24 | 72.37 | 56.11 |
| Forward T (Patrini et al., 2017) | 88.24 | 86.88 | 86.14 | 87.04 | 79.79 | 57.01 |
| Co-Teaching+ (Yu et al., 2019) | 90.61 | 89.70 | 89.47 | 89.54 | 83.26 | 57.88 |
| Peer Loss (Liu & Guo, 2020) | 90.75 | 89.06 | 88.76 | 88.57 | 82.00 | 57.59 |
| ELR+ (Liu et al., 2020a) | 94.83 | 94.43 | 94.34 | 94.34 | 91.09 | 66.72 |
| Positive-LS (Lukasik et al., 2020) | 91.57 | 89.80 | 89.35 | 89.82 | 82.76 | 55.84 |
| DivideMix (Li et al., 2020) | 95.01 | 95.16 | 95.23 | 95.21 | 92.56 | 71.13 |
| Negative-LS (Wei et al., 2021a) | 91.97 | 90.29 | 90.37 | 90.35 | 82.99 | 58.59 |
| CORES* (Cheng et al., 2020) | 95.25 | 94.45 | 94.45 | 94.87 | 91.66 | 55.72 |
| VolMinNet (Li et al., 2021b) | 89.70 | 88.30 | 88.27 | 88.53 | 80.53 | 57.80 |
| CAL (Zhu et al., 2021) | 91.97 | 90.93 | 90.75 | 90.74 | 85.36 | 61.73 |
| PES (Semi) (Bai et al., 2021) | 94.66 | 95.06 | 95.06 | 95.22 | 92.68 | 70.36 |
| **PMP(Ours)** | 95.78 | 96.14 | 95.81 | 96.16 | 95.16 | 74.59 |

## H. Experimental Setup for Scenario 3 (Backdoor Attacks)

### H.1. Networks and Parameters

To thoroughly validate the effectiveness of the proposed PMP, we employ commonly used network architectures in the backdoor attack scenario to obtain both contaminated and clean parameters for the experiments. Specifically, for generating contaminated parameters on the CIFAR-10 dataset, we use PreAct-ResNet-18 (He et al., 2016b) and VGG19-BN (Simonyan, 2014) as backbone networks. For the Tiny-ImageNet dataset, we use PreAct-ResNet-18 (He et al., 2016b) as the backbone network. During training, for CIFAR-10, we apply AutoAug (Cubuk et al., 2019) and Cutout (DeVries, 2017) as data augmentation strategies, while for Tiny-ImageNet, we use RandAugment (Cubuk et al., 2020). Additionally, we use the Adam optimizer with a weight decay of $5 \times 10^{-4}$, an initial learning rate of $1 \times 10^{-3}$ with a cosine annealing learning rate decay, a batch size of 128, and train for 300 epochs, followed by several rounds of fine-tuning to obtain more contaminated parameters. For clean parameters, we train on the CIFAR-10 and Tiny-ImageNet datasets without any backdoor attacks using the corresponding backbone networks and training settings, also fine-tuning for multiple rounds to obtain more clean parameters.

### H.2. Metrics

We use two common metrics to evaluate the model's performance under different backdoor attacks: accuracy on clean samples (ACC) and attack success rate (ASR) on poisoned samples.

## I. More Results for Scenario 1 (Imbalanced Samples)

As shown in Table 6, we performed PMP on more contaminated parameters caused by long-tail distributions and compared the results with conventional SOTA imbalanced learning methods. It is further evident that the proposed PMP method effectively purifies the contaminated parameters caused by imbalanced samples.

For example, as shown in Table 6, for ResNet-32 on the CIFAR-100-LT dataset with an imbalance factor of 100, after applying PMP to purify the contaminated parameters, the model's performance improved to 60.5, far surpassing the current best SOTA performance of 52.8. Moreover, for ResNet-32 on the CIFAR-10-LT dataset with an imbalance factor of 50, after applying PMP to purify the contaminated parameters, the model's performance improved to 91.2, also significantly exceeding the current best SOTA performance of 84.3.

## J. More Results for Scenario 2 (Inprecise Labels)

As shown in Table 7, Table 8, and Table 9, we conducted parameter purification experiments on contaminated parameters caused by noisy labels and compared the results with conventional state-of-the-art (SOTA) noisy label learning methods. It is further evident that the proposed PMP method effectively purifies the contaminated parameters caused by noisy labels.

For example, as shown in Table 7, for ResNet-34 on CIFAR-10 with synthetic asymmetric noise at a ratio of 40%, after

*Table 10.* Performance of PMP on contaminated parameters of PreAct-ResNet-18 caused by backdoor attacks in CIFAR-10.

| Attack | BadNets-A2O | | BadNets-A2A | | Blended | | Input-Aware | | LF | | SSBA | | Trojan | | WaNet | |
| --- | --- | --- | --- | --- | --- | --- | --- | --- | --- | --- | --- | --- | --- | --- | --- | --- |
| Metric | ACC | ASR | ACC | ASR | ACC | ASR | ACC | ASR | ACC | ASR | ACC | ASR | ACC | ASR | ACC | ASR |
| Backdoored | 91.50 | 94.91 | 91.87 | 74.86 | 93.29 | 99.66 | 90.43 | 97.52 | 93.51 | 98.91 | 92.92 | 97.74 | 93.32 | 1.00 | 90.87 | 97.27 |
| FP (Liu et al., 2018a) | 91.77 | 0.84 | 92.05 | 1.31 | 92.74 | 10.17 | 94.05 | 1.62 | 92.05 | 21.32 | 92.21 | 20.27 | 92.24 | 67.73 | 92.94 | 0.66 |
| NAD (Li et al., 2021d) | 88.82 | 1.96 | 90.73 | 1.61 | 92.25 | 47.64 | 94.08 | 0.92 | 91.72 | 75.47 | 92.15 | 70.77 | 92.18 | 5.77 | 93.07 | 0.73 |
| NC (Wang et al., 2019a) | 90.27 | 1.62 | 89.79 | 1.11 | 93.69 | 99.76 | 93.84 | 10.48 | 93.01 | 99.06 | 92.88 | 97.07 | 91.85 | 51.03 | 92.80 | 98.90 |
| ANP (Wu & Wang, 2021) | 91.65 | 3.83 | 92.33 | 2.56 | 93.45 | 47.14 | 94.06 | 1.57 | 92.53 | 26.38 | 92.02 | 16.18 | 92.71 | 84.82 | 93.24 | 1.54 |
| i-BAU (Zeng et al., 2021a) | 87.43 | 4.48 | 89.39 | 1.29 | 89.43 | 26.82 | 89.91 | 0.02 | 88.92 | 11.99 | 86.53 | 2.89 | 89.29 | 0.54 | 90.70 | 0.88 |
| EP (Zheng et al., 2022) | 89.80 | 1.26 | 88.72 | 3.00 | 91.94 | 48.22 | 93.68 | 2.88 | 91.97 | 84.73 | 91.67 | 4.33 | 92.32 | 2.49 | 90.47 | 96.52 |
| NPD (Zhu et al., 2024) | 88.93 | 1.26 | 91.41 | 0.89 | 91.18 | 0.41 | 89.57 | 0.11 | 90.06 | 0.21 | 90.88 | 2.77 | 92.37 | 6.51 | 91.57 | 0.80 |
| **PMP(Ours)** | 91.89 | 1.14 | 91.54 | 1.08 | 92.14 | 2.15 | 92.86 | 2.64 | 91.61 | 2.12 | 92.28 | 1.91 | 92.13 | 2.12 | 91.67 | 0.75 |

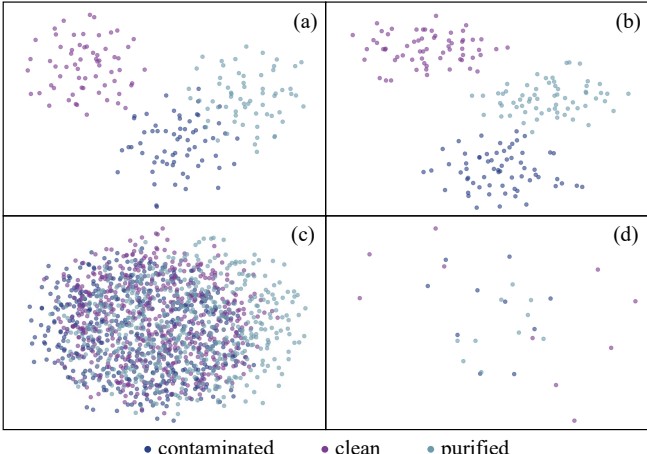

• contaminated    • clean    • purified

*Figure 5.* t-SNE visualization of contaminated, clean, and purified parameters. These parameters are extracted from a PreAct-ResNet-18 model trained on CIFAR-10, where each point represents a parameter cluster. (a) and (b) correspond to the first and second layers, while (c) and (d) correspond to the penultimate and final layers, respectively.

applying PMP to purify the contaminated parameters, the model's performance improved to 94.1, surpassing the performance of conventional SOTA noisy label methods.

Furthermore, as shown in Table 9, for ResNet-18 under real noise of the "Worst" type, after applying PMP to purify the contaminated parameters, the model's performance improved to 95.16, significantly exceeding the current best SOTA performance of 92.68.

## K. More Results for Scenario 3 (Backdoor Attacks)

As shown in Table 10, we conducted further parameter purification experiments on contaminated parameters caused by various backdoor attacks and compared the results with conventional SOTA backdoor attack defense methods. It is further evident that the proposed PMP method effectively purifies the contaminated parameters caused by backdoor attacks.

For example, as shown in Table 10, for PreAct-ResNet-18 under a WaNet backdoor attack on CIFAR-10, after applying PMP to purify the contaminated parameters, the model's performance on clean samples improved to 91.67, and the attack success rate on poisoned samples decreased to 0.75, surpassing the performance of conventional SOTA backdoor defense methods. Similarly, for the SSBA backdoor attack, after applying PMP to purify the contaminated parameters, the model's performance on clean samples improved to 92.28, and the attack success rate on poisoned samples decreased to 1.91, also exceeding the performance of conventional SOTA backdoor defense methods.

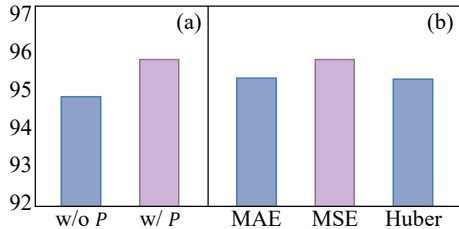

*Figure 6.* Impact of local manifold position information (subplot a) and various reconstruction losses (subplot b) on the accuracy of parameter manifold denoising. "w/ $P$" and "w/o $P$" represent the cases with and without the use of position information, respectively. "Huber" represents the Huber loss.

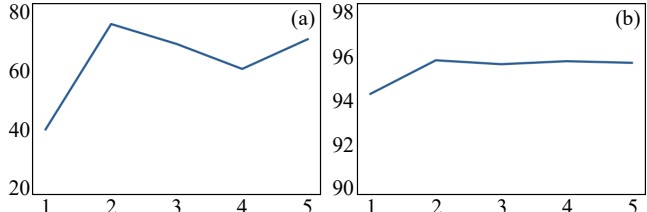

*Figure 7.* Impact of hyperparameter $b$ in local manifold positional information on the accuracy of implicit manifold learning (subfigure a) and implicit manifold denoising (subfigure b). The x-axis represents different values of $b$, while the y-axis represents the accuracy of reconstructed or denoised parameters.

## L. Visualization of Various Parameter Distributions

As shown in Fig. 5, we use t-SNE to visualize the distribution of contaminated parameters, clean parameters, and purified parameters within the same model. In the shallow layers of the model, the distribution of noisy contaminated parameters differs significantly from that of clean parameters. Moreover, the distribution of purified parameters differs from that of contaminated parameters, indicating that the proposed PMP effectively transforms the contaminated parameters into a new, distinct distribution of purified parameters, ultimately improving model accuracy. In contrast, in the deeper layers of the model, the distributions of contaminated, clean, and purified parameters become more intertwined. A potential reason for this phenomenon is that deeper layers are more closely related to high-level semantic features, making the differences between contaminated and clean parameters inherently smaller.

## M. Effectiveness of Different Reconstruction Losses in Implicit Manifold Denoising

As shown in Figure 6(b), we compared the impact of different reconstruction losses on the accuracy of implicit manifold denoising. It can be observed that the choice of loss function does not significantly affect the accuracy of the implicit manifold denoising. For example, the accuracy of implicit manifold denoising using MSE and MAE are 95.33 and 95.80, respectively. A potential reason for this is that the implicit manifold learning stage can learn a compact low-dimensional manifold that removes redundant information. Without the interference of redundant information, the Diffusion model is more capable of denoising. Moreover, by introducing a small amount of noise during the restoration of parameter cluster local manifolds, the denoising process becomes more robust, thereby reducing sensitivity to the denoised compact low-dimensional manifold. As a result, even ordinary reconstruction losses can work well for implicit manifold denoising in Diffusion models.

## N. Effect of Hyperparameter $b$ in Local Manifold Positional Information

As shown in Fig. 7, we investigate the impact of hyperparameter $b$ from Eq.(6) in the main text on the accuracy of implicit manifold learning (subfigure a) and implicit manifold denoising (subfigure b). It can be observed that implicit manifold learning is relatively sensitive to the choice of $b$, whereas implicit manifold denoising is less affected. For instance, in implicit manifold denoising, the accuracy of the denoised parameters remains nearly identical regardless of whether $b$ is set to 2 or 5. However, in implicit manifold learning, the highest accuracy for reconstructed parameters is achieved when $b = 2$. A potential reason for this phenomenon is that hyperparameter $b$ influences the fundamental frequency of positional encoding. Variations in the fundamental frequency affect the Fourier features of the final positional encoding. Since the multilayer

*Table 11.* Training and inference cost of PMP.

| Module | Manifold Learning | Manifold Denoising |
|---|---|---|
| Training Time | 19.7 h | 10.4 h |
| Training FLOPs | 2.0e17 FLOPs | 2.2e17 FLOPs |
| Training GPU Memory | 5605.6 MB | 4067.3 MB |
| Training CPU Memory | 1605.3 MB | 2821.7 MB |
| Inference Time | 1 s | 380.4 s |
| Inference FLOPs | 3.3e12 FLOPs | 4.6e15 FLOPs |
| Inference GPU Memory | 716.6 MB | 987.6 MB |
| Inference CPU Memory | 1570.1 MB | 1755.5 MB |
| GPU Model | NVIDIA RTX A6000 | NVIDIA RTX A6000 |
| Number of GPUs | 1 | 1 |

*Table 12.* Performance comparison between PMP and several representative weight-generation baselines on the parameter purification task. "(one layer)" denotes purifying only a single layer of the model. "-" indicates that the method cannot be directly adapted to the parameter purification task, and thus no results are reported.

| Method | Head | Medium | Tail | Overall |
|---|---|---|---|---|
| p-diff | 1.1 | 0.9 | 1.1 | 1.0 |
| p-diff (one layer) | 69.2 | 43.4 | 17.6 | 49.7 |
| D2NWG | 1.2 | 1.0 | 1.1 | 1.1 |
| D2NWG (one layer) | 69.3 | 43.6 | 17.7 | 49.9 |
| HyperDiffusion | - | - | - | - |
| **PMP (Ours)** | **85.1** | **84.1** | **80.2** | **82.6** |

perceptron $g_\phi$ has a relatively small number of parameters, it relies more heavily on the characteristics of positional encoding when fitting the high-dimensional parameter cluster manifold, particularly for capturing high-frequency variations.

## O. Comparison with Baseline Models

In addition to validating the effectiveness of the proposed PMP across multiple representative adverse scenarios, we also adapt several diffusion-based weitght-generation methods to the parameter purification task proposed in this paper. During experimentation, aside from differences in the method pipelines themselves, all training data and training configurations are kept identical. Furthermore, without altering the original design principles of these baseline methods, we carefully tune their configurations to ensure a fair comparison.

Taking the imbalanced-data scenario as an example, we conduct parameter purification experiments on the ViT-B/16 model, as shown in Table 12. The results indicate that existing representative methods are unable to effectively improve model performance when applied to parameter purification. When the number of parameters involved in purification is large, for example, purifying all parameters, these classical methods may even degrade model performance due to limited precision in the generated parameters.

## P. Training and Inference Efficiency of Parameter Manifold Purification

In addition to the performance comparisons across different scenarios, we also report the training and inference efficiency of PMP. All experiments are conducted on a single NVIDIA RTX A6000 GPU, requiring no expensive computational resources and thus being accessible to typical researchers. Using the noisy-label scenario as an example, the computational costs of PMP's manifold learning module and manifold denoising module are shown in Table 11. As indicated, both the training and inference costs of PMP fall within a highly acceptable range in terms of computation and time.

In particular, the inference cost demonstrates a significant advantage: while optimizing a model through ImageNet training typically requires tens to hundreds of GPU hours, a single round of PMP purification takes only a few hundred seconds—representing a difference of two orders of magnitude (and with DDIM (Song et al., 2020) accelerated sampling, inference can theoretically be even faster). Furthermore, PMP follows a "train once, reuse many times" paradigm. Once trained, PMP can be applied to purify models that were not seen during training, without requiring retraining for each contaminated model, highlighting its strong potential for high efficiency.

*Table 13.* Comparison between PMP and existing related methods in terms of scenario generality and architecture generality. ✓ and ×
indicate applicable and not applicable, respectively.

| Method | Imbalanced-sample scenario | Imprecise-label scenario | Backdoor-attack scenario | ResNet-family architectures | Transformer-family architectures |
|---|---|---|---|---|---|
| cRT | ✓ | × | × | ✓ | × |
| MiSLAS | ✓ | × | × | ✓ | × |
| LiVT | ✓ | × | × | × | ✓ |
| PEL | ✓ | × | × | ✓ | ✓ |
| Forward | × | ✓ | × | ✓ | × |
| ELR | × | ✓ | × | ✓ | × |
| DivideMix | × | ✓ | × | ✓ | × |
| ANP | × | × | ✓ | ✓ | × |
| PDB | × | × | ✓ | ✓ | ✓ |
| NPD | × | × | ✓ | ✓ | × |
| **PMP (Ours)** | ✓ | ✓ | ✓ | ✓ | ✓ |

## Q. Scope of PMP

PMP is described as a general, scenario-agnostic, and architecture-agnostic optimization paradigm in the sense that its
design is not tied to any single contamination scenario or model architecture. In contrast to many existing methods that
are typically developed for a specific scenario or a specific architecture, PMP provides a unified parameter purification
framework that can be applied across multiple performance-degradation scenarios and model architectures. This property is
supported by the multi-scenario and multi-architecture experiments reported in the paper.

To further clarify this point, Table 13 shows that PMP, as a unified framework, covers multiple types of performance-
degradation scenarios and multiple types of model architectures. The table is intended to illustrate the scope of scenarios and
architectures covered by PMP, rather than to serve as evidence that generalization to unseen scenarios or unseen architectures
has been fully established.

## R. Limitations

Although PMP demonstrates remarkable effectiveness, several limitations remain. At the current stage, PMP has only been
validated in classic adverse scenarios where model parameters undergo contamination, such as imbalanced data, noisy labels,
and backdoor attacks. Its applicability to other scenarios, such as parameter contamination caused by hardware interference
during model transmission or storage, or contamination introduced during model pruning or quantization, requires further
investigation in the future.

Moreover, PMP has so far been evaluated only on mainstream vision models. Whether it can be effectively applied to
language models, multimodal models, or even large-scale foundation models remains an open question. In addition, since
the primary contribution of this work is the introduction of the novel optimization paradigm of parameter purification, the
generalizability of the PMP method itself warrants deeper exploration. Due to limitations in time and scope, addressing
these issues will be a key focus of our future work.

