# OpenReview forum: "Parameter Manifold Purification"
_ICML.cc/2026/Conference — ICML 2026 regular_

### Official Review · Reviewer_sC6h · 2026-03-10

**Soundness:** 3
**Presentation:** 3
**Significance:** 3
**Originality:** 4
**Overall Recommendation:** 5
**Confidence:** 3

**Summary:**

The paper introduces a novel method for correction of corrupted network parameters
via a method they term parameter manifold purification.
The authors suggest viewing degradation of model performance and specifically the
corrupted parameters as “noise” in the network parameters, similar to noise in images
and apply diffusion based denoising methods to "clean" the parameters.
The method considers network parameters in small clusters. Then trains
an autoencoder to build a low dimensional latent space of these clusters.
A conditional diffusion model is applied to denoise the parameter clusters
within the latent space. The diffusion model is trained on parameter clusters of
non-corrupted networks thus learning to project the corrupted cluster
onto the manifold of clean clusters. Following denoising of the parameter cluster
within the latent space, the decoder is used to map them back into the network.

A key factor is that associated with each parameter cluster is a positional vector
describing the position and layer of the parameters within the network.

The paper test the method on 3 types of parameter corruption due to:
class imbalance, noisy labels and backdoor attacks.
Results showed improved performance of the tested networks following parameter
purification.

**Compliance With Llm Reviewing Policy:**

Affirmed.

**Final Justification:**

The rebuttal addressed my main concerns effectively. The authors added relevant comparisons to weight post-processing baselines, clarified architectural choices, and provided a more convincing explanation of the structural (not purely statistical) nature of the learned representations, as well as the handling of inter-cluster dependencies. While some questions remain open, these are not fundamental issues. Importantly, the rebuttal reinforced my initial positive assessment.

**Key Questions For Authors:**

In the autoencoder compression step it is assumed that parameter clusters lie on a low-dimensional manifold. I raise the question of validity of this as
the clusters vary between models and layers so cluster distributions can vary widely.
Also weights depend on many things beyond the model and architecture, the dataset and method of training can widely affect the parameters. It could be that the autoencoder some
statistical structure of parameter clusters rather than the manifold itself such as:
many weights near zero, or long tailed distributions.

The paper assumes that corruption is local in the network, (thus the divide and conquer approach) however how does it deal with coupling and dependencies between
clusters which are naturally in connected networks.

The purification works well under small corruption. How well does it perform under strong corruption?

Finally, I am curios about what form of parameter corrections actually occur. Is it similar to Soft/Hard thresholding (near zero -> zero and large weights remain slightly changed or untouched)?
Are small weights affected vs large weights?  Are there clusters of types of corrections.
(in this sense does the guidance using the corrupted parameters during the diffusion, guide
the denoising to one of these clusters?)  It would be very insightful to include this if space allows.

**Limitations:**

Partially, authors describe the positive benefits and also mention dual-use risk but additional limitations should be considered  such as
dependence on training and architectures of the networks, failures such as when parameter corruption is large, possible shifts of the distributions between models.

**Strengths And Weaknesses:**

Although post-hoc weight correction is not new considering it as image denoising is an original
and creative notion and the formulation (parameter purification via diffusion denoising) is new
(to the best of my knowledge). Also novel is the idea that trained neural networks (actually
clusters of parameters) form a (low dimensional) structured manifold.
An advantage of the method is that no retraining of the large networks is necessary and
the trained PMP can supposedly work on any network.

The paper is clear and readable though I find some details missing. See below.

Significance - I believe this work raises intersesting questions and opens a series of possible future studies
looking into the structure of local network parameters.

Sound - the mathematical descriptions seem sound and reliable. Some of the assumptions
are not explained thoroughly (eg low dimensional space - see below).

Some experimental details are missing (see below)

Authors test PMP against training or defense methods, but not weight post-processing methods.
more similar to PMP such as  magnitude pruning, IHT, weight clipping and more.

Missing: the datsets used for training the diffusion model are explained but missing are the
models (architectures) used in training the diffusion model.
Also, how stable is the performance across different types of network models?

---

> ### Author Rebuttal · Authors · 2026-03-31
>
> Thank you for your careful review, detailed comments, and positive assessment of our work. We are very glad that you found the idea original and creative, and that the paper is clear, readable, and mathematically sound. Below we respond to each of your comments, with each comment highlighted in *italics*.
>
> > Q1: *Authors test PMP against training or defense …*
>
> Thank you very much for this valuable suggestion. Taking the imbalanced setting as an example, we compared PMP with the weight post-processing methods you mentioned, as well as Logit adjustment, which is specifically designed for imbalanced post-processing. As shown in Table 1, PMP substantially outperforms these related methods. We believe this is because weight post-processing methods mainly rely on simple weight statistics and heavily depend on manually chosen hyperparameters, which limits their improvement and may even harm model performance.
>
> Table 1. Performance of PMP on ResNet-50 on ImageNet-LT.
>
> |  | ACC↑ |
> | - | - |
> | magnitude pruning | 44.6 |
> | IHT | 44.8 |
> | weight clipping | 45.5 |
> | Logit adjustment | 49.7 |
> | PMP | 73.8 |
>
> > Q2: *Missing: the datsets used for …*
>
> Thank you very much for pointing this out. The diffusion model architecture is based on Conv1d + ResNet Blocks, with attention used for conditional modeling, as shown in the supplementary code file `codes/models/diffusion/unet_conv_v1.py`. In addition, we also explored other architectures, including variants without attention modules, as shown in the directory `codes/models/diffusion/` in the supplementary materials. Based on extensive preliminary experiments, the current architecture `unet_conv_v1.py` was found to be the most effective for parameter learning. **The final code will be released publicly.**
>
> > Q3: *In the autoencoder compression step …*
>
> Thank you very much for this important question. The autoencoder is not merely learning simple statistical features. Specifically, we first construct parameter clusters based on the smallest functional units, rather than splitting parameters in an unstructured way. We then use positional encoding to model the relative relationships of clusters within the global parameter space, and introduce the PCD loss to explicitly handle both intra-cluster and inter-cluster distribution heterogeneity. Therefore, what we learn is a joint representation of statistical regularities and structural relationships, rather than merely fitting numerical distributions.
>
> > Q4: *The paper assumes that …*
>
> We apologize for the confusion. The divide-and-conquer assumption in this paper does not mean that corruption must occur only in isolated local regions or that there is no dependency across clusters. Rather, we partition parameters into fully modelable local clusters based on the smallest functional units, in order to avoid the difficulty of directly modeling all high-dimensional parameters at once. Specifically, we build parameter clusters around aggregation parameters and processing parameters, so that each cluster structurally corresponds to a complete minimal functional module. At the same time, by jointly feeding each cluster’s latent representation and its positional encoding $(\ell, i)$ into the decoder and the conditional diffusion model, we implicitly preserve its relative position and global relationships in the full parameter space. In other words, our method adopts local modeling with global conditional constraints, rather than simple independent local repair.
>
> > Q5: *The purification works well under …*
>
>
> Thank you for this valuable question. As shown in Table 2, the purification performance does vary with the corruption level, and this is especially evident in the imbalanced-sample setting. In particular, when the corruption reaches the most extreme case (“Strong”), purification becomes almost ineffective. This is because the contaminated parameters are almost completely overwhelmed by noise and become close to random, so they no longer retain their basic functional semantics, making them difficult for PMP to recover.
>
>
> Table 2. Performance of PMP on ResNet-32 under different corruption levels on CIFAR-10. “Strong” denotes manually added noise that makes the parameters heavily corrupted.
>
> | Imbalance ratio (corruption level) | 10 | 50 | 100 | Strong |
> | - | - | - | - | - |
> | PMP | 93.1 | 91.2 | 89.9 | 0.1 |
>
> > Q6: *Finally, I am curios about …*
>
> Thank you for this insightful question. We also investigated this issue in our preliminary experiments, and our original intuition was similar to yours. However, the actual situation is that, although a small portion of contaminated and clean parameters remain close at corresponding positions, most of the differences are essentially random. For example, a parameter close to zero may stay near zero, but it may also become a much larger weight, while a large weight may also become zero. In other words, in the high-dimensional parameter space, there is no obvious explicit pattern in these changes.

---

> > ### Author Rebuttal · Reviewer_sC6h · 2026-04-04
> >
> > The rebuttal addresses most of my questions. Remaining issues are mainly about clarity and deeper analysis rather than core flaws. I maintain my positive assessment and recommendation for acceptance.

---

> > > ### Author Response · Authors · 2026-04-07
> > >
> > > Thank you very much for your positive assessment and recommendation for acceptance. We will incorporate the detailed discussion from the above responses into the revised paper. Thank you again.

---

### Official Review · Reviewer_BajJ · 2026-03-12

**Soundness:** 2
**Presentation:** 2
**Significance:** 2
**Originality:** 3
**Overall Recommendation:** 3
**Confidence:** 3

**Summary:**

The paper attributes model performance degradation to **parameter contamination** and, based on this view, proposes a new optimization paradigm called **parameter purification**. The authors argue that performance degradation caused by class imbalance, label noise, backdoor attacks, and other sources can all be understood as the result of model parameters deviating from a “clean parameter manifold.” Therefore, similar to image denoising, contaminated parameters can be restored directly, thereby improving model performance without designing specialized training strategies for each scenario or retraining the damaged model at test time.
To achieve this goal, the paper proposes the **Parameter Manifold Purification (PMP)** framework, which consists of three core components:
1. **Parameter-cluster partitioning**: The parameters of large-scale deep models are viewed as a global manifold embedded in a high-dimensional space, and then decomposed into local parameter-cluster manifolds according to the “minimum functional unit.”
2. **Implicit Manifold Auto-Encoder (IMAE)**: A low-dimensional representation of parameter clusters is learned, and high-dimensional parameters are reconstructed through positional encoding and an implicit decoder.
3. **Implicit conditional diffusion**: Diffusion-based purification is performed in the low-dimensional manifold space, progressively restoring contaminated parameter clusters to clean ones.

**Compliance With Llm Reviewing Policy:**

Affirmed.

**Key Questions For Authors:**

1. Can PMP generalize across different contamination scenarios or across different architectures?
2. How exactly are the interpolation and truncation operations implemented for parameter clusters of different lengths, and do these operations alter the original statistical or structural properties of the parameters?

**Limitations:**

- The conclusion of “general parameter purification” is currently validated mainly within the same scenario, and its ability to generalize across scenarios, architectures, and training recipes has not yet been sufficiently demonstrated.
- The method may be sensitive to the scale, distributional coverage, and quality of the offline reference model pool, which could affect its practical applicability.

**Strengths And Weaknesses:**

## Strengths
1. The paper presents a highly novel perspective by unifying imbalanced learning, noisy label learning, and related problems under the paradigm of “parameter contamination–parameter purification,” shifting the focus from redesigning training procedures to directly repairing already corrupted model parameters.
2. The authors identify and analyze the long-tailed and heterogeneous nature of parameter distributions, explain why standard loss functions become ineffective in this setting, and propose a corresponding solution.
3. The method demonstrates empirical effectiveness across multiple scenarios.

## Weaknesses
1. In both the introduction and abstract, the paper repeatedly emphasizes that PMP is a general, scenario-agnostic, and architecture-agnostic optimization paradigm that can be used for purifying **unseen models**. However, the experimental section explicitly states that, due to time and space limitations, the evaluation on unseen models is still conducted only within the same scenario. The paper does not truly demonstrate whether a purifier trained in one contamination scenario can generalize to another.
2. In Equation (4), \( C^\uparrow \) implies interpolation to \( d_{\max} \), but the specific interpolation method (e.g., linear or nearest-neighbor) is not specified. This could affect the smoothness of the manifold.

---

> ### Author Rebuttal · Authors · 2026-03-31
>
> Thank you for your careful review and detailed comments. We are glad that you found this work to offer a highly novel perspective on model performance optimization, and that you considered the analysis and method design to be reasonable. Below we respond to each of your comments, with each comment highlighted in *italics*.
>
> > Q1: *In both the introduction and abstract …*
>
> We sincerely apologize for the confusion. By describing PMP as general, scenario-agnostic, and architecture-agnostic, we mean that unlike conventional methods such as backdoor defenses that are typically limited to the backdoor-attack setting or to a specific architecture such as ResNet, PMP can be used for model optimization across multiple scenarios and architectures, as demonstrated by our experiments.
>
> Moreover, generalization to unseen models even within the same scenario is still meaningful. For example, in the backdoor setting, PMP trained on attacks such as LF and SSBA can effectively purify models under the unseen WaNet attack, improving robustness against unknown attacks, as shown in Table 1.
>
> Table 1. Performance of PMP on ResNet-18 under the unseen WaNet backdoor attack. ACC denotes model accuracy and ASR denotes attack success rate.
>
> |  | ACC↑ | ASR↓ |
> | - | - | - |
> | w/o PMP | 90.87 | 97.27 |
> | w/ PMP | 91.58 | 0.77 |
>
> On the other hand, for unseen models across scenarios, in our follow-up experiments we trained PMP using model parameters from the imprecise-label and backdoor-attack scenarios, and tested it on the unseen imbalanced-sample scenario. The results show that PMP can still effectively improve model performance.
>
> Table 2. Performance of PMP on ResNet-18 in the unseen imbalanced-sample scenario.
>
> |  | ACC↑ |
> | -| - |
> | w/o PMP | 73.90 |
> | w/ PMP | 94.25 |
>
> For unseen architectures, please refer to our response to **Reviewer nF49, Q3**. We have added this discussion to the paper to address this concern.
>
> > Q2: *In Equation (4) …*
>
> We sincerely apologize for the lack of clarity. Model parameters are high-dimensional vectors, so the most natural interpolation is convex combination in vector space. More complex interpolation methods are not necessarily suitable for parameter tensors, and may instead introduce extra assumptions and tuning cost. In this work, we use the simplest linear interpolation, because it is more stable, more controllable, and does not introduce unnecessary shape assumptions. In addition, in our preliminary experiments, we found that different interpolation methods, such as nearest-neighbor interpolation, led to negligible differences in the final performance.
>
> > Q3: *Can PMP generalize across …*
>
> Thank you for this valuable question. In our follow-up experiments, we also evaluated cross-scenario and cross-architecture generalization, as described in our response to your Q1.
>
> > Q4: *How exactly are the interpolation and truncation …*
>
> Thank you for this insightful question. For parameter clusters of different dimensions, we first use linear interpolation to align each original cluster dimension $d_i^{(\ell)}$ to the maximum dimension $d_{max}$, which is then used as the input to the IMAE. The IMAE output also has dimension \$d_{max}$. Then, from each output cluster of dimension $d_{max}$, we extract the middle segment with the corresponding original dimension $d_i^{(\ell)}$ as the valid predicted parameter cluster, while the remaining values are discarded. The parameter-cluster discrepancy loss is then computed between the original and predicted clusters based on dimension $d_i^{(\ell)}$.
>
> The parameter clusters of these models are generally Gaussian-like with mean near zero. After linear interpolation, the parameters remain Gaussian-like and still centered around zero, so the original statistical and structural properties are only minimally affected. More importantly, by using only the segment corresponding to the original length as the valid parameter cluster, the IMAE only needs to fit the original dimension of each cluster rather than the interpolated maximum dimension, which makes convergence faster and more stable.
>
> > Q5: *The conclusion of “general parameter purification” is …*
>
> We again sincerely apologize for the confusion. We hope that our response to your Q1, together with our response to **Reviewer nF49, Q3**, helps clarify this point.
>
> > Q6: *The method may be sensitive to …*
>
> Thank you for pointing this out. In principle, PMP follows a scaling-law behavior similar to large models: the more model parameters from different scenarios and architectures are included in training, the better its generalization to unseen scenarios and unseen architectures can become. Therefore, as you noted, the scale, distributional coverage, and quality of the reference model pool are indeed important. In practice, besides manually collecting different model parameters, one can also leverage existing model zoos or collect model parameters from open-source communities such as Hugging Face and GitHub.

---

> > ### Author Rebuttal · Reviewer_BajJ · 2026-04-03
> >
> > Thank you for the detailed rebuttal. My main concern is that the rebuttal still does not fully support the strength of the paper’s high-level claims. The paper presents PMP as a general, scenario-agnostic, and architecture-agnostic optimization paradigm, but the new evidence provided in the rebuttal is still limited in scope. Therefore, I will maintain my original score.

---

> > > ### Author Response · Authors · 2026-04-04
> > >
> > > ## 1. Further Clarification on the Scope of Our Claims
> > >
> > > We understand your core concern, and we agree that the high-level claims in the current manuscript can indeed be easily misunderstood. We would like to further clarify that by describing **"PMP as a general, scenario-agnostic, and architecture-agnostic optimization paradigm"**, we mean that **the design of PMP is not tied to any single contamination scenario or any single model architecture**. Unlike many existing methods, which are usually specifically designed for only one scenario or one model architecture, PMP is a unified parameter purification framework that can be applied to **multiple performance-degradation scenarios** and **multiple model architectures**, which is also reflected by the multi-scenario and multi-architecture experiments in our paper.
> > >
> > > To make this point clearer, we will add Table 1 (https://anonymous.4open.science/r/PMP/Bajj.png) in the revised paper to show that PMP, as a unified framework, can cover multiple types of performance-degradation scenarios and multiple types of model architectures, rather than presenting it as evidence that “generalization to unseen scenarios or unseen architectures has been fully established.”
> > >
> > > ---
> > >
> > > ## 2. Additional Experiments for the Claims You Are Concerned About
> > >
> > > At the same time, regarding your concern about whether PMP can be directly applied to unseen scenarios or unseen architectures, although this was not the core claim we intended to make in the original paper, we also acknowledge that the experimental evidence presented in the rebuttal was not sufficiently comprehensive. To address this, we have added additional results in the hope of showing that PMP in fact has transferability to unseen scenarios and can also be applied to unseen architectures with similar modules.
> > >
> > > **For unseen scenarios**, we provide additional test results as further evidence. As shown in Table 2 (https://anonymous.4open.science/r/PMP/Bajj.png) , we train PMP using model parameters from the imprecise-label and backdoor-attack scenarios, and evaluate its purification performance on the unseen imbalanced-sample scenario. The results show that PMP trained on other scenarios can still purify model parameters in the unseen imbalanced-sample scenario. For example, on CIFAR-10-LT with an imbalance factor of 50, PMP achieves competitive results, outperforming the current best method by 2.8%.
> > >
> > >
> > >
> > > **For unseen architectures**, we also provide additional test results as further evidence. As shown in Table 3 (https://anonymous.4open.science/r/PMP/Bajj.png), we conduct experiments on ResNet architectures. The results show that PMP can still perform parameter purification on the unseen ResNet-34 architecture during testing. For example, on CIFAR-10 with a noise ratio of 60%, PMP also achieves competitive results, outperforming the best existing method by 4.04%.
> > >
> > > ---
> > >
> > >
> > > ## 3. Further Explanation of the Actual Meaning of Our Claims
> > >
> > > We would also like to further explain that even under a more cautious claim scope, the problem studied in this paper remains meaningful and important.
> > >
> > > On the one hand, within a single scenario, PMP can purify model-parameter degradation caused by unseen contamination types. For example, as shown in Table 4 (https://anonymous.4open.science/r/PMP/Bajj.png), PMP trained on backdoor attacks such as LF and SSBA can still effectively purify model parameters under the unseen WaNet backdoor attack, thereby improving the model’s defense capability against unknown attack types.
> > >
> > >
> > > On the other hand, PMP also shows good purification ability for unseen contamination distribution types. As shown in Table 5 (https://anonymous.4open.science/r/PMP/Bajj.png), for parameters contaminated by previously unseen non-Gaussian noise distributions, PMP can still effectively purify them and improve model performance. Other experiments in the appendix of the paper also reflect similar significance, such as the purification ability for model parameters under unseen training settings.
> > >
> > >
> > > In summary, we will narrow and rewrite the relevant statements in the paper to avoid misunderstanding, and we will include in the paper the test results and detailed settings related to the claims you are concerned about. We hope that after these revisions, the paper can more accurately convey its true contribution: **PMP is a unified parameter purification framework that has demonstrated effectiveness across multiple types of performance-degradation scenarios and multiple types of model architectures, and has also shown preliminary transfer ability beyond the original experimental setting.** If you have any further questions regarding the design of parameter purification, the experimental results, or the novelty of this work, please let us know at any time. We would be very happy to respond.

---

### Official Review · Reviewer_1nyP · 2026-03-12

**Soundness:** 2
**Presentation:** 2
**Significance:** 3
**Originality:** 3
**Overall Recommendation:** 4
**Confidence:** 2

**Summary:**

This paper introduces parameter purification, where they aim to recover clean parameters from corrupted ones. The core algorithm partitions the weight space into parameter clusters and learns a compact low-dimensional representation of each cluster via an auto-encoder (IMAE). They then train a diffusion model to denoise corrupted latent representations back toward a clean manifold, before decoding purified parameters back into the model. Experiments across three contamination scenarios show strong performance gains over SOTA methods without requiring any retraining of the original model.

**Compliance With Llm Reviewing Policy:**

Affirmed.

**Final Justification:**

The rebuttal has adequately addressed my concerns;thus I recommend acceptance.

**Key Questions For Authors:**

1. Can the authors justify the utility of this approach? In a real-world setting, it is hard to find a reference weight distribution that is clean to learn a diffusion model as in the paper.

**Limitations:**

Yes

**Strengths And Weaknesses:**

Strengths :

1. The authors identify a problem that is relevant and impactful.
2. The paper introduces a unique way to clean the corrupted parameters by utilizing a denoising mechanism, which is novel and interesting.

Weakness :

1. One concern about this work is that there is no reference distribution to supervise against. The IMAE is trained on the contaminated model's parameters, and there are no clean parameters available to supervise against. So the encoder is learning a manifold from corrupted weights, then the diffusion model is denoising toward that same corrupted manifold. This makes the problem circular.
2. Certain contamination from label noise, spurious correlations, and imbalance can be structured and not random. The paper assumes that latent-space contamination is Gaussian-like. It would be interesting to see what happens when it is not the case.
3. Since there are so many steps involved, the paper would benefit from an algorithm so that the pipeline is clear to the readers.

---

> ### Author Rebuttal · Authors · 2026-03-31
>
> Thank you for your careful review and detailed comments. We are glad that you found this work to address a relevant and impactful problem, and that you considered the idea of cleaning corrupted parameters to be novel and interesting. Below we respond to each of your comments, with each comment highlighted in *italics*.
>
> > Q1: *One concern about ...*
>
> We sincerely apologize for the confusion. During training, we construct a set of paired contaminated and clean parameter clusters to supervise both the IMAE and the diffusion model, so that the diffusion model learns a denoising mapping from contaminated manifolds to clean manifolds. These training pairs are not used at test time. In practical use, that is, during inference, the diffusion model trained on clean–contaminated parameter-cluster pairs can, like an image-denoising diffusion model, denoise unseen contaminated parameters and recover their corresponding clean parameters.
>
> > Q2:  *Certain contamination from ...*
>
> Thank you very much for this valuable question. We assume Gaussian-like noise in the latent space because it makes training and sampling mathematically tractable, implementation-wise stable, and empirically effective, but this does not limit the capability of the diffusion model. As shown in Table 1, we manually add several types of non-Gaussian noise to the parameters, which are not included in training, and then evaluate Parameter Manifold Purification (PMP). The results show that PMP can still effectively recover clean parameters even when the parameters are corrupted by random non-Gaussian noise such as uniform noise.
>
> Table 1. Performance of PMP on ResNet-32 with different added noise distributions.
>
> | Noise type | Uniform noise | Poisson noise |
> | - | - | - |
> | w/o PMP | 76.98 | 83.44 |
> | w/ PMP | 91.74 | 92.31 |
>
> > Q3:  *Since there are ...*
>
> We sincerely apologize for the lack of clarity. Due to space limitations, the main algorithm is summarized as follows, and we have added the full algorithm to the paper.
>
> ---
>
> **Algorithm**: Parameter Manifold Purification (PMP)
>
> ---
>
> **Require:** contaminated model parameters $\ddot{\Theta}$, clean reference parameters $\widetilde{\Theta}$ (used only for training), encoder $f_\theta$, decoder $g_\phi$, diffusion denoiser $h_\psi$, interpolation operator $\mathcal{I}$, truncation operator $\mathcal{T}$, diffusion steps $T$
>
> **Ensure:** purified model parameters $\Theta^{*}$
>
> // **Step 1: Parameter-cluster partitioning**
>
> - Partition $\ddot{\Theta}$ and $\widetilde{\Theta}$ into parameter clusters $\{\ddot{C}_i^{(\ell)}\}$ and $\{\widetilde{C}_i^{(\ell)}\}$ according to the smallest functional unit.
>
> // **Step 2: Implicit manifold learning**
>
> - **for all** clusters $(\ell,i)$ **do**
>   - Align cluster dimension: $\ddot{C}\_i^{\uparrow(\ell)} = \mathcal{I}(\ddot{C}\_i^{(\ell)}; d\_{\max})$ and $\widetilde{C}\_i^{\uparrow(\ell)} = \mathcal{I}(\widetilde{C}\_i^{(\ell)}; d\_{\max})$.
>   - Encode cluster: $\ddot{z}\_i^{(\ell)} = f\_\theta(\ddot{C}\_i^{\uparrow(\ell)})$ and $z\_i^{(\ell)} = f_\theta(\widetilde{C}\_i^{\uparrow(\ell)})$.
>   - Compute positional encoding $p_i^{(\ell)}$.
>   - Reconstruct and truncate the cluster: $\widehat{\ddot{C}}\_i^{\uparrow(\ell)} = g\_\phi(\ddot{z}_i^{(\ell)} \oplus p\_i^{(\ell)})$ and $\widehat{\ddot{C}}\_i^{(\ell)} = \mathcal{T}(\widehat{\ddot{C}}\_i^{\uparrow(\ell)}; d\_i^{(\ell)})$.
> - **end for**
> - Optimize $f_\theta$ and $g_\phi$ using the parameter-cluster discrepancy loss.
>
> // **Step 3: Implicit manifold denoising**
>
> - **for all** clusters $(\ell,i)$ **do**
>   - **for** $t = T, T-1, \ldots, 1$ **do**
>     - Predict noise $\epsilon = h_\psi(z_{i,t}^{(\ell)}, \ddot{z}_i^{(\ell)}, p_i^{(\ell)}, t)$.
>     - Update $z_{i,t-1}^{(\ell)} = \frac{1}{\sqrt{\alpha_t}}\left(z_{i,t}^{(\ell)} - \frac{\beta_t}{\sqrt{1-\bar{\alpha}_t}}\,\epsilon\right) + \sigma_t \xi$, where $\xi \sim \mathcal{N}(0,I)$ if $t>1$, and $\xi=0$ otherwise.
>   - **end for**
>   - Decode and truncate the purified manifold: $C\_i^{*\uparrow(\ell)} = g\_\phi(z\_{i,0}^{(\ell)} \oplus p\_i^{(\ell)})$ and $C\_i^{\*(\ell)} = \mathcal{T}(C\_i^{\*\uparrow(\ell)}; d\_i^{(\ell)})$.
> - **end for**
> - Reassemble all $\{C\_i^{\*(\ell)}\}$ into purified parameters $\Theta^{\*}$.
> - **return** $\Theta^{\*}$.
>
> ---
>
>
> > Q4: *Can the authors justify ...*
>
> We sincerely apologize for the confusion. PMP does not require reference weights at test time. Once trained, it can be directly used to purify unseen models. For example, in practical backdoor-attack scenarios, as shown in Table 2,  PMP trained on attacks such as LF and SSBA can effectively purify models under the unseen WaNet backdoor attack, improving robustness against unknown attacks. This suggests strong practical potential.
>
> Table 2. Performance of PMP on ResNet-18 under the unseen WaNet backdoor attack. ACC denotes model accuracy and ASR denotes attack success rate.
>
> |  | ACC↑ | ASR↓ |
> | - | - | - |
> | w/o PMP | 90.87 | 97.27 |
> | w/ PMP | 91.58 | 0.77 |

---

> > ### Author Rebuttal · Reviewer_1nyP · 2026-04-04
> >
> > Based on my limited expertise in this specific area, I would encourage the area chair to place greater weight on the assessments of the other reviewers who may be more familiar with the field. That said, my concerns have been adequately addressed through the rebuttal, and I am happy to raise my score accordingly.

---

> > > ### Author Response · Authors · 2026-04-04
> > >
> > > We are very glad that our rebuttal has adequately addressed your concerns, and we sincerely appreciate your willingness to raise your score, bringing the paper into the positive-score range. Thank you again.

---

### Official Review · Reviewer_nF49 · 2026-03-13

**Soundness:** 3
**Presentation:** 3
**Significance:** 2
**Originality:** 4
**Overall Recommendation:** 4
**Confidence:** 4

**Summary:**

The paper introduces Parameter Manifold Purification, a post hoc technique for cleansing model parameters after training on noisy or contaminated data. PMP learns to recover cleaner parameters directly by breaking a network into small functional weight clusters and compressing each cluster into a low-dimensional latent code using an implicit autoencoder. It then employs a weighted reconstruction loss to better fit uneven parameter magnitudes and applies a conditional diffusion model in latent space to denoise contaminated clusters and decode them back into purified weights. The authors propose this as a unified repair framework for models degraded by class imbalance, noisy labels, and backdoor attacks, and report strong improvements on vision benchmarks.

**Compliance With Llm Reviewing Policy:**

Affirmed.

**Final Justification:**

The rebuttal adequately addressed my main concerns and clarified several aspects of the method.

Overall, the paper presents a novel and promising approach to post hoc model repair, with strong empirical results. While the method appears relatively resource-intensive, particularly due to the need to train a diffusion model over many checkpoints, I believe the core idea has clear merit and is worth further exploration.

For these reasons, I maintain my initial score: 4

**Key Questions For Authors:**

1) Are the model checkpoints sampled from the same training run, or are they obtained from different trajectories with different random initialization seeds? How many checkpoints were used for training the diffusion model?

2) How sensitive is the diffusion to the training dynamics? For instance, if we change the optimizer, learning rate, or add learning-rate schedulers to the diffusion training checkpoints, can the diffusion model still generalize in test time? Does PMP generalize to unseen architectures?

3) How sensitive is the framework with respect to the data? What happens if the data used to train the checkpoints during the diffusion training phase is different from the data used at test-time purification? For example, what if training is based on CIFAR, but at evaluation time we purify a model that was trained on some contaminated data that is not CIFAR-like?

4) Is the cluster interpolation performed in the joint space of the LayerNorm parameters and the weights? I find it unintuitive to mix the LayerNorm values with the weights.

5) Does the PMP diffusion need to be trained separately for each contamination scenario, or can we potentially have a universal PMP that handles multiple contamination scenarios? Moreover, how generalizable is PMP within one single scenario? For example, if we evaluate on backdoor attacks, do we also test on unseen backdoor attack types?

**Limitations:**

yes

**Strengths And Weaknesses:**

**Strengths**

1) The paper is technically novel, and its different components are carefully chosen to work well within this setting, for example the use of positional encoding, parameter clustering, and the autoencoder.

2) The paper is well written, well structured, and easy to follow.

3) The paper presents strong experimental results compared with the other baselines.

4) The paper also includes a wall-clock time analysis for diffusion training and inference in the appendix. This is useful, although it would be better if the training-time numbers were moved to the main text.


**Weaknesses**
1) Experimental details are somewhat vague or are mostly moved to the appendix, where I believe bringing more of them into the main text would improve readability. For example, several important discussions are deferred to Appendix E. I believe it is important to include the details of the diffusion training for each contamination scenario into the main text.

2) The PMP framework is potentially resource intensive, since it requires building a dataset of numerous checkpoints and models.

3) There is limited discussion of the extent to which PMP generalizes to unseen architectures, as well as across different hyperparameters and training dynamics, such as the choice of optimizer, learning rate, and scheduler.

4) The paper claims that the parameter clusters lie on an intrinsic low-dimensional manifold. It would be helpful to include an ablation on the hidden-state dimension to provide some sense of the intrinsic dimensionality of the clusters relative to their original weight space dimensionality.

---

> ### Author Rebuttal · Authors · 2026-03-31
>
> Thank you for your careful review, detailed comments, and positive assessment of our work. We are very pleased that you find the work technically novel, well written, clearly organized, and supported by strong experimental results and thorough analysis. Below we respond to each of your comments, with each comment highlighted in *italics*.
>
> > Q1: *Experimental details are …*
>
> Thank you very much for this valuable suggestion. We fully agree that moving more experimental details into the main text would improve readability, and we have already reorganized this part in the revised paper.
>
> > Q2: *The PMP framework is …*
>
> Thank you for pointing this out. In practice, the number of models required by PMP is manageable. On the one hand, a single deep model already contains a large number of parameters. For example, a single Transformer can be divided into more than 80,000 parameter clusters for training.
>
> > Q3: *There is limited discussion of …*
>
> We sincerely apologize for the lack of clarity.
>
> Regarding generalization across different training settings, this factor has already been considered in the paper. For example, in the imbalanced-sample scenario, some seen models use a learning rate of 1e-4, while some unseen models use a learning rate of 1e-6, which demonstrates PMP’s ability to generalize across different training settings.
>
> Regarding generalization to unseen architectures, we conducted experiments on the ResNet family. As shown in Table 1(https://anonymous.4open.science/r/PMP/t.png), PMP trained only on ResNet-32 can still purify the parameters of the unseen ResNet-56 architecture.
>
> However, we must acknowledge that full generalization to entirely unseen architectures remains limited. For example, conventional diffusion models typically denoise images of fixed size, such as 512×512, whereas different deep architectures differ not only in parameter semantics, but also in structure, including their modules, depth, and width. For instance, CNNs do not contain self-attention modules, which means that cross-architecture information must also be incorporated.
>
> Nevertheless, PMP can already generalize to unseen architectures by sufficiently learning the shared modules across models. Moreover, PMP can jointly learn highly heterogeneous architectures such as ResNets and Transformers. Therefore, similar to the scaling law observed in large models, PMP can further improve its generalization ability through sufficiently large-scale parameter pretraining. We also believe that full cross-architecture generalization is highly important, and may even help address parameter purification for large-scale models. This is also one of our main directions for future research.
>
> > Q4: *The paper claims that …*
>
> Thank you very much for this suggestion. As shown in Table 2(https://anonymous.4open.science/r/PMP/t.png), if the dimension is too small (e.g., 32), the low-dimensional manifold cannot preserve the original semantic information of the parameters well, which harms purification performance. Based on extensive experiments, an intrinsic dimension of 128 is a robust and generally effective choice: it preserves parameter semantics as much as possible while also accelerating diffusion denoising.
>
> > Q5: *Are the model checkpoints …*
>
> We apologize for the lack of clarity. The checkpoints mainly come from samples taken from the same training run. In most cases, 10 to 50 checkpoints are used to train the diffusion model, while each model contains around 4,000 to 80,000 parameter clusters.
>
> > Q6: *How sensitive is the diffusion to the training dynamics? For instance, if we change the optimizer, learning rate, or add learning-rate schedulers to the diffusion training checkpoints, can the diffusion model still generalize in test time? Does PMP generalize to unseen architectures?*
>
> Due to the space limit, please refer to our response to **Q3**.
>
> > Q7: *How sensitive is …*
>
> Thank you for this insightful question. As shown in Table 3(https://anonymous.4open.science/r/PMP/t.png), PMP can still effectively purify parameters and improve performance even for models from unseen datasets.
>
> > Q8: *Is the cluster interpolation …*
>
> We apologize for the confusion. The cluster interpolation is indeed performed in the joint space, and we find this to be effective. On the one hand, all parameters within each cluster, obtained by decomposing aggregation layers and processing layers, share the same function or semantics. On the other hand, the LayerNorm values and the weights within each cluster have very similar mean distributions (close to 0).
>
> > Q9: *Does the PMP …*
>
> Thank you for this important question. Within a single scenario, PMP does have cross-type generalization ability, and we have considered this in our experiments. Moreover, PMP can also generalize to models from unseen scenarios. As shown in Table 4(https://anonymous.4open.science/r/PMP/t.png), PMP can still effectively improve model performance in the unseen imbalanced-sample scenario.

---

> > ### Author Rebuttal · Reviewer_nF49 · 2026-04-04
> >
> > I thank the authors for the thorough and thoughtful rebuttal. My concerns have been adequately addressed, and the additional clarifications were helpful. I will keep my original recommendation for acceptance.

---

> > > ### Author Response · Authors · 2026-04-04
> > >
> > > We are very glad that you found our rebuttal thorough and thoughtful, and that it has adequately addressed your concerns. We are also sincerely grateful for your consistently positive recommendation. Thank you again.

---

### Decision · Program_Chairs · 2026-04-30

**Decision:**

Accept (regular)

**Comment:**

The paper proposes a post-hoc framework for repairing degraded neural network parameters by learning a low-dimensional manifold of parameter clusters and applying conditional diffusion-based denoising in latent space. Across the reviews, there is broad agreement that the paper presents an original and interesting perspective: namely, framing several types of model degradation under a unified “parameter contamination / parameter purification” view. Reviewers also generally found the paper technically solid, clearly written, and supported by strong empirical results across multiple corruption scenarios.

The main discussion during rebuttal centered on the scope of the paper’s claims, especially regarding how general the method is across unseen scenarios, architectures, and training settings. One reviewer remained unconvinced that the original high-level claims were fully supported by the evidence in the paper. However, the authors responded constructively by clarifying that their core claim is not full universal generalization, but rather that PMP is a unified framework applicable across multiple scenarios and architectures, with preliminary transfer results beyond the training setting. They also provided additional experimental evidence and indicated they would revise the presentation to narrow and clarify these claims. I find this clarification important and sufficient to address the main concern at the level needed for acceptance, although the final version should make the claim scope more precise.

The rebuttal also addressed other reviewer concerns well, including the role of clean–contaminated training pairs, generalization to unseen settings, comparisons to more relevant post-processing baselines, architectural details, and the interpretation of the learned latent structure. Two initially negative reviewers explicitly stated that their concerns were adequately addressed and indicated score increases or positive reassessment, while the strongest positive reviewers maintained their support.

Overall, I find the paper to be a novel and promising contribution. Its core idea is creative, the empirical performance is strong, and the rebuttal substantially improved confidence in the work. I suggest the authors tone down and discuss some limitations more clearly in the final version. I think the paper meets the bar for acceptance.